# Atmospheric odd nitrogen response to electron forcing from a 6D magnetospheric hybrid-kinetic simulation

Tuomas Häkkilä[1], Maxime Grandin[2,1], Markus Battarbee[2], Monika E. Szeląg[1], Markku Alho[2], Leo Kotipalo[2], Niilo Kalakoski[1], Pekka T. Verronen[1,3], and Minna Palmroth[1,2]

[1]Space and Earth Observation Centre, Finnish Meteorological Institute, Helsinki, Finland
[2]Department of Physics, University of Helsinki, Helsinki, Finland
[3]Sodankylä Geophysical Observatory, University of Oulu, Sodankylä, Finland

**Correspondence:** Tuomas Häkkilä (tuomas.hakkila@fmi.fi)

**Abstract.** Modelling the distribution of odd nitrogen ($NO_x$) in the polar middle and upper atmosphere has proven to be a complex task. Firstly, its production by energetic electron precipitation is highly variable across a range of temporal scales from seconds to decades. Secondly, there are uncertainties in the measurement-based but simplified electron flux data sets that are currently used in atmosphere and climate models. The altitude distribution of $NO_x$ is strongly affected by atmospheric dynamics also on monthly time scales, particularly in the polar winter periods when the isolated air inside the polar vortex descends from lower thermosphere to mesosphere and stratosphere. Recent comparisons between measurements and simulations have revealed strong differences in the $NO_x$ distribution, with questions remaining about the representation of both production and transport in models. Here we present for the first time a novel approach, where the electron atmospheric forcing in the auroral energy range ($50\,\mathrm{eV} - 50\,\mathrm{keV}$) is derived from a magnetospheric hybrid-kinetic simulation with a detailed description of energy range and resolution, and spatial and diurnal distribution. These electron data are used as input in a global whole atmosphere model to study the impact on polar $NO_x$ and ozone. We show that the magnetospheric electron data provide a realistic representation of the forcing which leads to considerable impact in the lower thermosphere, mesosphere and stratosphere. We find that during the polar winter the simulated auroral electron precipitation increases the polar $NO_x$ concentrations up to $215\,\%$, $59\,\%$, and $7.8\,\%$ in the lower thermosphere, mesosphere, and upper stratosphere, respectively, when compared to no auroral electron forcing in the atmospheric model. These results demonstrate the potential of combining magnetospheric and atmospheric simulations for detailed studies of solar wind – atmosphere coupling.

## 1 Introduction

Gaining insight into the polar mesosphere–lower thermosphere–ionosphere (MLTI) region poses a particular challenge. The MLTI extends from roughly 80 to 200 km, a range that is beyond the reach of many ground-based observations and below the optimal range for most satellite measurements (Palmroth et al., 2021). Therefore, accurate modelling of the MLTI is essential

to augment the scarce direct measurements and deepen our understanding of this region. The polar MLTI (polewards of 60°) is influenced by solar radiation, which generates both daily and seasonal variations due to the planet's rotation and axial tilt. Additionally, this region is also affected by the electromagnetic forces from the magnetosphere, leading to complex interactions between the neutral atmosphere and the ionosphere. These interactions include several mechanisms concerning the energetics, dynamics, and chemistry of the MLTI that remain poorly understood (Sarris et al., 2023). While these topics are intricately linked, this study focuses on the chemistry part, particularly on the role of energetic particle precipitation on the ionisation and composition of the MLTI. More specifically, we focus on the role of auroral electrons in impacting the composition of the polar MLTI in atmospheric modelling.

Nitric Oxide (NO), a member of the odd nitrogen family ($NO_x$, defined as the sum of N, NO, and $NO_2$), is one of the most important species in the polar MLTI energetics (e.g. Mlynczak et al., 2005). Through its descent inside the polar vortex, $NO_x$ also provides a dynamical connection between MLTI and stratospheric altitudes (Funke et al., 2014). In the upper stratosphere, $NO_x$ transport from above leads to depletion of ozone which has been measured using satellite-based instruments (Damiani et al., 2016). Because ozone strongly absorbs solar ultraviolet radiation, it is one of the main constituents determining the thermal structure of the stratosphere. Through ozone, $NO_x$ can have an impact on the radiative balance of the polar atmosphere beyond just within the MLTI.

In the atmosphere, $NO_x$ is produced primarily as the result of solar radiation. However, in the polar regions, especially during the polar night, energetic particle precipitation (EPP) is a major driver of $NO_x$ production (Barth et al., 2001). There are three primary EPP sources of $NO_x$: solar proton events, radiation belt electrons, and auroral electrons (Verronen and Rodger, 2015). The energy of solar protons and radiation belt electrons are large enough for them to penetrate and produce $NO_x$ in the mesosphere and stratosphere, while auroral electrons, with typical energies on the order of a few kiloelectronvolts, are limited to thermospheric altitudes. Auroral precipitation occurs on a continuous basis into the upper atmosphere in the polar regions, particularly along the auroral oval, located most of the time between 60° and 75° geomagnetic latitude. The auroral electron flux is significantly enhanced during magnetospheric substorms, when the magnetotail is suddenly disrupted and launches a large number of electrons (and protons) of variable energies towards the ionosphere (Palmroth et al., 2017; Palmroth et al., 2023). Substorms and pulsating aurora, a phenomenon during the substorm recovery phase, have been shown to produce $NO_x$ (Seppälä et al., 2015; Turunen et al., 2016). In fact, auroral electrons are probably the the largest contributor to the overall polar MLTI $NO_x$ budget (Sinnhuber et al., 2011).

Atmospheric models struggle to produce correct amounts of $NO_x$ in the MLTI when compared to observations (Randall et al., 2015), leading to an incomplete representation of the radiative balance within the polar region through the $NO_x$ impact on stratospheric ozone (Szeląg et al., 2022). Explanations for the discrepancy in $NO_x$ production vary. Previous studies have shown that the transport of thermospheric $NO_x$ to mesospheric and stratospheric altitudes remains a challenge in models (Meraner and Schmidt, 2016; Smith-Johnsen et al., 2022). Hendrickx et al. (2018) also suggested that models underestimate the radiation belt electron forcing, and that models have inadequate ion chemistry schemes. The lack of specific focus on auroral electron precipitation may also be a contributing factor to the $NO_x$ underestimation. In long-term climate simulations, models typically include the $NO_x$ production from auroral electrons using proxy models based on geomagnetic indices, and typically

use a simplistic representation for their energy spectrum (e.g. Marsh et al., 2007). Detailed electron models that include auroral energies exist (e.g. Wissing and Kallenrode, 2009) and have been used in atmospheric model comparisons (Nesse Tyssøy et al., 2022; Sinnhuber et al., 2021), but emphasis is often on electrons at medium energies (30–1000 keV).

The objective of this paper is to present the first results of a new method to quantify the chemical response of the upper and middle atmosphere to auroral electron precipitation. We report on the first-time combination of magnetospheric and atmospheric modelling, and show impacts on $NO_x$ and ozone concentrations resulting from auroral electron forcing. We use eVlasiator, a variant of the global hybrid-Vlasov model Vlasiator, to simulate the electron fluxes at auroral energies from 50 eV to 50 keV. Atmospheric ionisation rates derived from these fluxes are then used in the Whole Atmosphere Community Climate
Model (WACCM) to determine the polar atmospheric $NO_x$ and $O_3$ impacts from the MLTI to the upper stratosphere. This study should be understood as an initial effort towards improving the description of the atmospheric effects of particle precipitation by including first principles in the modelling of the auroral electron fluxes, rather than relying on empirical parametrisation of the fluxes. The modelled magnetospheric electron fluxes characterise the altitude extent and distribution of the forcing in more detail than proxy-based parametrisations. We compare the $NO_x$ impact of auroral electron fluxes derived from eVlasiator to
that of WACCM's internal auroral forcing. Since current atmospheric models struggle to produce enough $NO_x$ in the MLTI, we study whether the more accurate description of auroral electrons from eVlasiator leads to an increased production of $NO_x$ compared to the simplistic internal auroral forcing in WACCM. We also discuss the limitations of our current approach as well as the potential in understanding the solar wind – atmosphere coupling.

## 2   Methods

### 2.1   Vlasiator and eVlasiator

Vlasiator is a global hybrid-Vlasov model simulating ion-kinetic plasma physics of near-Earth space (Palmroth et al., 2018), which recently became capable of running 6D (three spatial dimensions, three velocity dimensions) simulations (Ganse et al., 2023). Vlasiator models the collisionless ion populations directly as velocity distribution functions (VDFs), discretised on Cartesian grids, allowing for accurate representation of phenomena such as wave-particle interactions (Dubart et al., 2020)
and precipitating protons (Grandin et al., 2019b, 2020, 2023), which cannot be modelled using the magnetohydrodynamic (MHD) codes (Palmroth et al., 2006). The spatial simulation domain is divided into either a uniform Cartesian 2D spatial mesh or a Cartesian 3D mesh with regions of interest refined with an octree cell-based refinement algorithm (Ganse et al., 2023; Kotipalo et al., 2024). Each spatial cell contains a 3D velocity mesh consisting of cubic uniform Cartesian cells. In order to fit the massive amount of simulation data into memory, Vlasiator utilises a sparse algorithm (von Alfthan et al., 2014) where
only those regions of velocity space which are deemed to contribute to plasma moments in a significant fashion are stored and propagated. This is implemented through discarding blocks of the grid which have phase-space density below a pre-defined threshold, yet maintaining a buffer region around those cells in order to ensure physical behaviour at the edges of the velocity domain. The simulation state is propagated directly via the Vlasov equation, with electric and magnetic fields solved on a

regular Cartesian grid and closure provided by MHD Ohm's law with the Hall and electron pressure gradient terms included (Palmroth et al., 2018).

A typical Vlasiator simulation models the global geomagnetic domain of the Earth, spanning tens to hundreds of Earth radii ($R_E = 6371$ km) in each dimension with a spatial resolution of the order of the ion inertial length, and with an inner boundary positioned at roughly 5 Earth radii. The velocity grid is defined so to be able to discretise the inflowing solar wind. The Earth's magnetic field is modelled as a standard dipole field with the dipole moment set to that of the actual Earth value, facilitating direct comparison with spacecraft observations. Sample simulation parametrisations can be found for example in Horaites et al. (2023), Palmroth et al. (2023), and Grandin et al. (2024). Vlasiator runs are propagated on the order of hundreds to thousands of seconds, in order to facilitate self-consistent formation of the magnetospheric domain and its dynamics, but constrained by the availability of computational resources.

eVlasiator is an offshoot of Vlasiator which considers electrons as a kinetic population (Battarbee et al., 2021) instead of the usual ions. eVlasiator is not a standard full-kinetic plasma code, instead evaluating electron response to ion-scale structures and fields. What eVlasiator does provide is realistic electron VDFs evaluated for a single point in time from a larger Vlasiator simulation, such as presented in Alho et al. (2022) and validated against spacecraft observations. Since Alho et al. (2022), eVlasiator has been extended to work on 6D Vlasiator inputs, with the code available via Zenodo (Pfau-Kempf et al., 2022). Thus, eVlasiator can be used to infer kinetic electron VDFs along field lines, and thus also allowing the calculation of precipitating electron fluxes. Due to numerical constraints, eVlasiator supports use of a reduced mass ratio, for example $m_p/m_e \approx 183.6$, $m_e/m_{e,\mathrm{phys}} = 10$ in Alho et al. (2022) and $m_p/m_e = 40$, $m_e/m_{e,\mathrm{phys}} = 45.9$ in this study. The use of heavier electrons allows for completing a simulation with significantly reduced computational resources.

## 2.2 Simulating precipitating particle fluxes

In (e)Vlasiator, precipitating particle differential number fluxes are calculated in every ordinary space cell, at every output time step in the simulation. At a given position $\mathbf{r}$ in ordinary space, the precipitating electron differential number flux $\mathcal{F}_e$ value at energy $E$ is given by

$$\mathcal{F}_e(E, \mathbf{r}) = \frac{v^2}{m_e} \langle f_e(\mathbf{r}, v, \theta, \varphi) \rangle_{\theta_0} \tag{1}$$

with $v = \sqrt{2E/m_e}$ the corresponding electron speed, $m_e$ the electron mass, $f_e$ the electron phase-space density, $\theta$ the pitch angle, $\varphi$ the gyrophase angle, and $\theta_0$ the bounce loss cone angle, and where $\langle . \rangle_{\theta_0}$ denotes averaging over $\theta$ and $\varphi$ inside the loss cone. The full derivation of the version of Eq. (1) for proton fluxes can be found in Grandin et al. (2019b). Subsequent studies investigating dayside and nightside auroral proton precipitation under various driving conditions are presented in Grandin et al. (2020, 2023) and Horaites et al. (2023).

In this study, we present for the first time precipitating electron fluxes obtained with eVlasiator. The Vlasiator run used as the basis for the eVlasiator simulation is the same as described in e.g. Palmroth et al. (2023), and the eVlasiator run is the first 3D-3V magnetospheric eVlasiator simulation. The Vlasiator simulation is driven by a constant solar wind of $V_x = -750\,\mathrm{km\,s^{-1}}$ with a density of $n_p = 10^6\,\mathrm{m^{-3}}$ and a temperature of $0.5\,\mathrm{MK}$. These driving conditions, while not being extremely frequent,

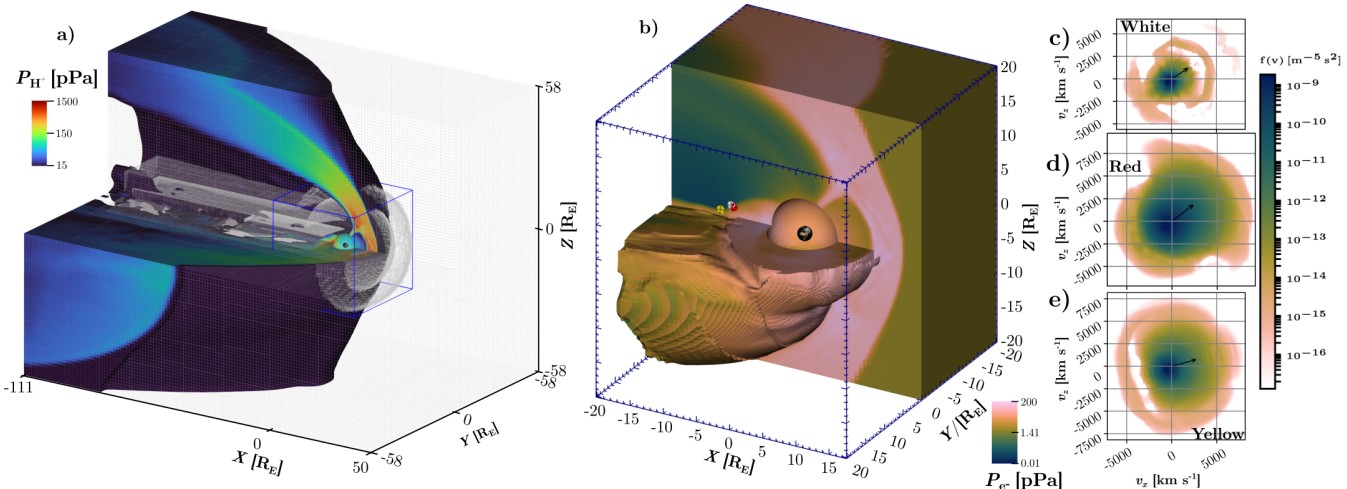

**Figure 1.** (a) Overview of the 3D-3V magnetospheric Vlasiator simulation, grid refinement regions (gray grids) and the interior of the magnetosphere, with the extracted eVlasiator domain shown as a blue box. Proton pressure is shown on the surfaces. (b) Overview of the eVlasiator simulation at its final state, with electron pressure shown on the bow shock, magnetopause, and the southern lobe. Earth is visible inside the spherical inner boundary of the simulation domain. (c–e) Examples of electron VDFs from eVlasiator on the midnight meridian, from the white, red, and yellow markers in panel b, showing diverse distribution functions and field-aligned beams on field lines connected to precipitation regions.

are comparable to those associated with the fastest of the solar wind high-speed streams (HSSs). HSSs reaching a velocity greater than $700 \, \mathrm{km \, s^{-1}}$ occur a few times per year, especially during the maximum and declining phases of the solar cycle (e.g. Grandin et al., 2019a). The inner boundary consists of stationary plasma at a radius of $4.7 \, R_\mathrm{E}$ and is modelled as a near-
conducting sphere. The spatial mesh has a base resolution of $8000 \, \mathrm{km}$ at the lowest refinement level, increasing up to $1000 \, \mathrm{km}$ in regions of interest such as the magnetotail and the magnetopause. The ion velocity cell resolution is $40 \, \mathrm{km \, s^1}$. The Vlasia-tor simulation was propagated for a total of 1506 seconds. The eVlasiator run based on the final state of the magnetospheric Vlasiator simulation utilised a mass ratio of $m_p/m_e = 40$ and an electron velocity cell resolution of $128 \, \mathrm{km \, s^{-1}}$, whilst main-taining the spatial resolution and fields of the Vlasiator simulation. Due to computational constraints, the eVlasiator simulation
was run selectively on the inner magnetosphere only, spanning $X \in [-20.1, 17.6] \, R_\mathrm{E}$ and $Y, Z \in [-20.1, +20.1] \, R_\mathrm{E}$, and was propagated for a time extent of $1.4 \, \mathrm{s}$. Figure 1 shows the Vlasiator and eVlasiator simulation domains and examples of electron velocity distributions in eVlasiator.

The eVlasiator output for WACCM modelling is the electron differential number flux (Eq. 1). Due to the eVlasiator constraint of a reduced mass ratio, the energisation of the high-mass electrons is assumed to be representative of that affecting real-mass

electrons in nature. Consequently, the eVlasiator differential number flux obtained with high-mass electrons in Vlasiator is estimated to be representative of the differential number flux of real-mass electrons, and is hence passed forward as-is for the construction of the forcing dataset.

## 2.3    Construction of the auroral-electron forcing dataset

### 2.3.1    Mapping of the ionosphere grid to the eVlasiator simulation domain

To obtain the forcing dataset for WACCM consisting of precipitating electron fluxes at auroral energies (0.05–50 keV) as a function of magnetic local time (MLT) and geomagnetic latitude (MLAT), we first construct a MLT–MLAT grid at ionospheric altitudes, which we map to the magnetosphere. The procedure is similar to that presented in Grandin et al. (2023), with a notable difference to account for the specific setup of the eVlasiator run. It is described in detail in Appendix A.

    It is worth highlighting that the obtained precipitating electron fluxes are the result of the physical processes described in the
eVlasiator simulation. These do not include some of the important processes for auroral electron acceleration and pitch-angle scattering, such as field-aligned potential drops above the ionosphere or inner-magnetospheric waves. As a consequence, the precipitating fluxes extracted from the eVlasiator run might differ from reality. For this reason, scaling with observational data has been performed as explained in the next section.

### 2.3.2    Scaling of the eVlasiator fluxes with DMSP observations

To scale the eVlasiator differential number fluxes with observations, we use two pairs of overpasses of Defense Meteorological Satellite Program (DMSP) spacecraft during similar driving conditions. Each pair contains a polar cusp overpass in the northern hemisphere (NH) and a nightside oval overpass in the southern hemisphere (SH). The dates of the events are 1 August 2011 (06:00–07:30 UT) and 10 October 2015 (05:30–06:30 UT). We use measurements from the Special Sensor J (SSJ) instrument (Redmon et al., 2017), which provides particle counts within a field of view spanning $4° \times 90°$ in the observation plane
(Hardy et al., 2008). Precipitating electron differential number fluxes are collected in 19 logarithmically spaced energy bins between 30 eV and 30 keV. For the first (second) event, SSJ measurements from the DMSP-F16 (F17) and DMSP-F18 (F17) spacecraft are considered. These two events were previously used for precipitating proton flux comparison between Vlasiator and observations in Grandin et al. (2023).

    A detailed description of the eVlasiator flux scaling based on the DMSP observations is given in Appendix B. In summary,
we apply an energy-dependent correction factor to the eVlasiator fluxes in the polar cusp and in the nightside auroral oval. There is one such correcting function for the cusp fluxes, $\alpha_{\mathrm{day}}(E)$, and another one for the nightside fluxes, $\alpha_{\mathrm{night}}(E)$. We first calculate the ratio between the measured differential number fluxes by DMSP/SSJ and those obtained with eVlasiator, along the DMSP orbit, separately for the dayside and nightside overpasses. Then, for a given electron energy, we examine the values taken by the obtained ratios along the DMSP orbits, and retain a certain percentile, QX = Xth percentile of along-orbit
eVlasiator-over-DMSP differential number flux. We obtain a curve of QX as a function of the precipitating electron energy, which we fit in logarithmic scale with a third-order (for the dayside) or second-order (for the nightside) polynomial function.

The choice of percentile X to consider for QX is a free parameter in the adjustment, which we constrain by aiming for obtaining a ratio between the integrated energy fluxes (corrected eVlasiator over DMSP) as close to one as possible (see Appendix B4 for details). We have determined that the 61st (67th) percentile optimally fulfils this condition for the dayside (nightside) ratios. Note that, to calculate the integrated energy flux of eVlasiator precipitation, we use the real electron mass, as the modified mass used for the simulation is cancelled out in the expression of the differential number flux (see Eq. (1)). Hence, no further mass correction is needed to infer the integrated energy flux. Finally, once the scaling ratios $\alpha_{\mathrm{day}}(E)$ and $\alpha_{\mathrm{night}}(E)$ are determined, we use them to calculate the corrected eVlasiator differential number fluxes in the corresponding region (dayside or nightside). This is done by multiplying the original fluxes by the ratios. Note that, while the fitting procedure is performed in log–log scales, the correction coefficients are indeed applied in the linear domain.

## 2.4   Whole Atmosphere Community Climate Model

The Whole Atmosphere Community Climate Model (WACCM) is a global 3D chemistry-climate model that covers the altitude range from the surface up to about $140\,\mathrm{km}$. The model incorporates various physical processes and interactions within the atmosphere, including dynamics, chemistry, radiation, and their interactions with the Earth's surface and external forcings such as solar radiation and greenhouse gases (Marsh et al., 2013; Gettelman et al., 2019). Here, we use WACCM-D, a variant of WACCM that enhances standard parameterisations of $HO_x$ and $NO_x$ production by incorporating a comprehensive ionospheric chemistry. This alteration aims to better replicate the observed impacts of energetic particle precipitation on the composition of the mesosphere and upper stratosphere (Verronen et al., 2016; Andersson et al., 2016). We conducted specified dynamics simulations (SD) where horizontal winds, temperature, pressure, surface stress and heat fluxes are adjusted to 3 hourly Modern-Era Retrospective Analysis for Research and Applications (MERRA-2) reanalysis data (Molod et al., 2015). The model is constrained by the reanalysis data up to about $50\,\mathrm{km}$ while the dynamics are free-running at altitudes above. We use version 6 of the model with a latitude $\times$ longitude resolution of $0.95° \times 1.25°$.

### 2.4.1   Energetic particle forcing in WACCM and implementation of auroral electrons from eVlasiator

In WACCM-D, ionisation by EPP drives the initial production rates of ions and neutrals due to particle impact ionisation, dissociative ionisation, and secondary electron dissociation (Verronen et al., 2016, Table 1). These rates are incorporated in the WACCM ion and neutral chemistry scheme, connecting EPP forcing to the $NO_x$ and ozone concentrations. As a default, WACCM input of solar and geomagnetic forcing is taken as recommended for the Coupled Model Intercomparison Project Phase 6 (CMIP6) (Matthes et al., 2017). In addition to total and spectral irradiance, this data set also includes atmospheric ionisation rates due to solar protons, medium-energy electrons, and galactic cosmic rays. These CMIP6 particle forcings are input into WACCM as daily atmospheric ion production rates. The solar proton forcing is based on satellite observations of proton fluxes at energies $1$–$300\,\mathrm{MeV}$ (Matthes et al., 2017), while the medium-energy electron forcing uses the electron precipitation model by van de Kamp et al. (2016) for energies $30$–$1000\,\mathrm{keV}$. We have included these recommended CMIP6 solar and geomagnetic forcing data in all our WACCM-D simulations.

Unlike the solar protons, medium-energy electrons, and galactic cosmis rays, WACCM's auroral electron forcing is not directly input as ionisation rates. Instead, the auroral electron precipitation forcing is driven by the daily geomagnetic $Kp$ index, based on the auroral model by Roble and Ridley (1987). The ionisation from auroral electrons is represented by a Maxwellian energy distribution and a characteristic energy of $2\,\mathrm{keV}$. WACCM also makes use of the three-dimensional nitric oxide empirical model (NOEM) to set NO concentration at WACCM's upper boundary. The inclusion of NOEM in WACCM simulations accounts for the production of NO above WACCM's altitude range. NOEM is driven by $Kp$, day of year, and solar $10.7\,\mathrm{cm}$ radio flux (Marsh et al., 2004). As such, NOEM also includes effects of auroral electrons on $NO_x$. However, since the auroral electrons mostly precipitate in the lower thermosphere (95–120 km) (e.g. Matthes et al., 2017), the main impact of auroral electrons falls well within WACCM's altitude range. For this reason we focus on replacing the default $Kp$-driven auroral model by Roble and Ridley (1987) with auroral forcing from eVlasiator in our WACCM simulations, and maintain NOEM as part of the WACCM setup.

In order to replace the default parametrisation of the auroral electron forcing within WACCM, a new ion production rate (IPR) input code was applied (Häkkilä, 2024). The new IPR code turns off the $Kp$-driven auroral model, and enables inputting auroral electron forcing as ionisation rates, similar to the other energetic particle forcing inputs. Since the eVlasiator auroral electron fluxes were available on a magnetic local time dependent grid, we implemented this as part of the new IPR code, enabling MLAT × MLT ionisation forcing. In addition to geomagnetic latitude, support was included for L-shell × MLT grids, as well as multiple time steps per day. A separate version of the IPR code which does not turn off the $Kp$ aurora was also created for possible future use.

Using the eVlasiator electron energy–flux spectra, we calculated corresponding forcing for our WACCM atmospheric simulations. We made use of the method of parameterised electron impact ionisation by Fang et al. (2010). This is the same method that was used to create electron ionisation rates for the CMIP6 (van de Kamp et al., 2016; Matthes et al., 2017). Here, however, the electron flux data are not from a proxy model based on satellite observations but from the eVlasiator magnetospheric simulations. The ionisation rate calculation requires an atmosphere which was taken from the NRLMSISE-00 model by Picone et al. (2002). To ensure consistency with the WACCM atmosphere, and in accordance with the CMIP6 procedure (Matthes et al., 2017), the ionisation rates are then divided by the NRLMSISE-00 mass density. When the rates are used as input, they are multiplied by the WACCM atmospheric mass density profiles by the new IPR code.

Since the method by Fang et al. (2010) is derived for precipitating electrons at energies $> 100\,\mathrm{eV}$ we limited the energy spectrum of the auroral electron fluxes from eVlasiator accordingly. We also limited the higher end of the eVlasiator electron energy spectrum: Since the CMIP6 medium-energy electron precipitation already accounts for electrons at energies $> 30\,\mathrm{keV}$, we removed energies 30–50 keV from the eVlasiator-derived electron flux energy range to prevent possible double counting. Thus, though the auroral electron fluxes were obtained from eVlasiator on an energy range from $50\,\mathrm{eV}$ to $50\,\mathrm{keV}$ with 32 logarithmically spaced individual grid points, the final ionisation rate calculation was performed at energies $100\,\mathrm{eV}$–$30\,\mathrm{keV}$. The atmospheric ionisation rates were calculated at magnetic latitudes between $63°$ and $88°$ in both hemispheres with one degree spacing. A half-an-hour resolution was used for the magnetic local time throughout the day. Figure 2 shows an example of eVlasiator spectra and corresponding atmospheric ionisation rates. Large differences in fluxes at different geomagnetic

latitudes result in a similarly large range of ionisation. According to the spectral energy range (electron energies $< 30\,\mathrm{keV}$), the bulk ionisation is restricted to altitudes above $90\,\mathrm{km}$.

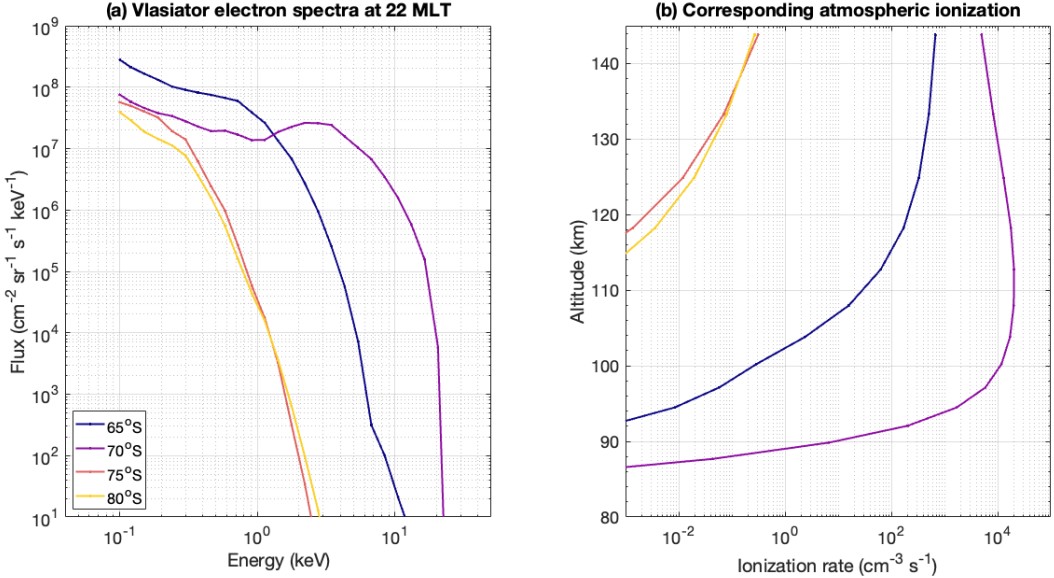

**Figure 2.** (a) eVlasiator electron spectra at 22 hours of magnetic local time at four southern hemisphere magnetic latitudes at energies 0.1–$30\,\mathrm{keV}$. (b) Corresponding atmospheric ionisation rates.

### 2.4.2 WACCM simulation setup and output

WACCM-D was run from January 2005 to June 2006 in order to cover both southern and northern hemispheric winters. In this paper we consider outputs of daily average and instantaneous auroral ionisation rates, as well as daily average $NO_x$ and ozone concentrations. The main WACCM-D simulation of this study with the eVlasiator-derived ionisation rates as auroral electron forcing (VLAS) was performed using the new IPR code for WACCM described in Sect. 2.4.1. We input the auroral ionisation rates in WACCM-D on a MLAT $\times$ MLT grid of $1° \times 0.5\,\mathrm{h}$ resolution. For each day of the simulation we use the same ionisation data, since only one time step was available from the eVlasiator output. Additionally, a reference run (REF) was performed using the new IPR code, with the ionisation input from auroral electrons set to 0 at all gridpoints as well as the $Kp$-driven aurora being turned off.

For comparisons with the eVlasiator auroral electron forcing, simulations were carried out using WACCM's parametrised $Kp$-driven aurora. Since the VLAS run was performed using the same data every day, we use a fixed $Kp$ value for the comparison runs. We performed 6 separate WACCM-D runs, each with a different, fixed $Kp$ index value from 0 to 5 (KP0–KP5), but we present here mostly the runs KP1–KP3, since those most closely correspond to the level of auroral ionisation and

impact found in the VLAS case. It should be noted that WACCM's default auroral electron forcing is such that using $Kp = 0$ does not result in no auroral forcing (see Fig. 4b), making the KP0 and REF runs materially different.

As stated in Sect. 2.4.1, WACCM uses NOEM to set the NO concentration at the model upper boundary during simulations. Since NOEM is driven by the $Kp$ index, this makes it necessary to also fix $Kp$ indices for the VLAS and REF runs to ensure comparability with the KP simulations. The $Kp$ value was fixed to 0 for the REF run to create minimal conditions for comparisons, and to 2 for the VLAS run. The choice of $Kp = 2$ for VLAS was made considering the DMSP overpasses used for the scaling of the eVlasiator electron fluxes, which took place during $Kp$ index values around 2 and 3, for 1 August 2011 and 10 October 2015, respectively. All the WACCM-D simulations presented here and their differences are given in Table 1.

**Table 1.** The WACCM-D simulation runs and differences in their setups.

| simulation | description | auroral forcing | $Kp$ index (fixed) |
|:---:|:---:|:---:|:---:|
| REF | Reference run with no aurora | none | 0 |
| KP0 | Default WACCM-D run with fixed $Kp = 0$ | parameterised | 0 |
| KP1 | Default WACCM-D run with fixed $Kp = 1$ | parameterised | 1 |
| **VLAS** | **Main run with eVlasiator auroral electrons** | **eVlasiator** | **2** |
| KP2 | Default WACCM-D run with fixed $Kp = 2$ | parameterised | 2 |
| KP3 | Default WACCM-D run with fixed $Kp = 3$ | parameterised | 3 |

## 3 Results

### 3.1 Auroral electron precipitation

#### 3.1.1 Auroral electron fluxes from eVlasiator

Figure 3 shows the integrated parameters of the auroral electron precipitation forcing dataset obtained from eVlasiator, after the scaling with DMSP/SSJ observations (see Sect. 2.3.2). Each panel gives the data as a function of geomagnetic latitude (radial coordinate) and MLT (angular coordinate). Figures 3a and 3b show the precipitating electron integrated energy flux in the northern and southern hemispheres, respectively. We can identify the cusp region on the dayside, between 70° and 80° MLAT and within 9–15 MLT, with flux magnitudes on the order of $10^8 \, \text{keV} \, \text{cm}^{-2} \, \text{s}^{-1} \, \text{sr}^{-1}$. On the nightside, the integrated energy flux peaks in the pre-midnight sector and within 65–70° MLAT, reaching magnitudes on the order of $10^9 \, \text{keV} \, \text{cm}^{-2} \, \text{s}^{-1} \, \text{sr}^{-1}$. The forcing is very symmetrical on the nightside, whereas slight differences can be noted in the polar cusps – these results are similar to those obtained for auroral proton precipitation and discussed in Grandin et al. (2023). Figures 3c and 3d present the mean precipitating electron energy. Values are below 1 keV on the dayside and reach up to 5 keV on the nightside.

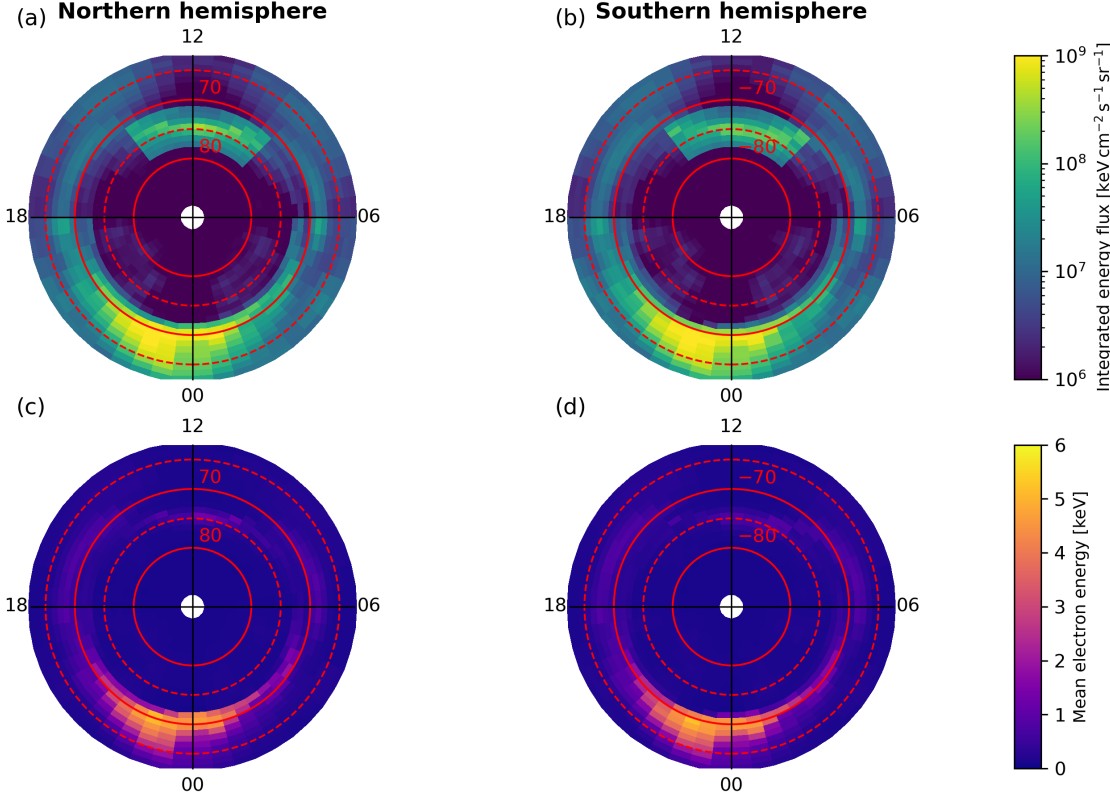

**Figure 3.** Polar view of integrated parameters of auroral electron precipitation in the eVlasiator run, after scaling with DMSP/SSJ observations (see Sect. 2.3.2). (a–b) Precipitating electron integrated energy flux in the northern and southern hemispheres. (c–d) Mean precipitating electron energy in the northern and southern hemisphere. In each panel, the radial coordinate is geomagnetic latitude, and the angular coordinate is magnetic local time.

### 3.1.2 Ionisation rates

Comparisons of the ionisation rates from eVlasiator and the $Kp$ parametrisation are shown in Figs. 4–6 for the northern hemisphere. The vertically integrated ionisation rates in Fig. 4 show the full auroral oval on geographic coordinates on 1 Jan 2006; panels a–e depict daily average ionisation, and panels f–j show instantaneous ionisation at 00 UT. Figures 5–6 show the ionisation along the geographic longitude 36.25°W, which has the maximal instantaneous eVlasiator-derived auroral ionisation rates at 00 UT.

In the integrated daily average ionisation rates (Fig. 4a–e) the eVlasiator-derived auroral ionisation rates roughly match the location of the $Kp$-driven ionisation rates, but the $Kp$ parametrisation has a wider spread within the oval than the eVlasiator

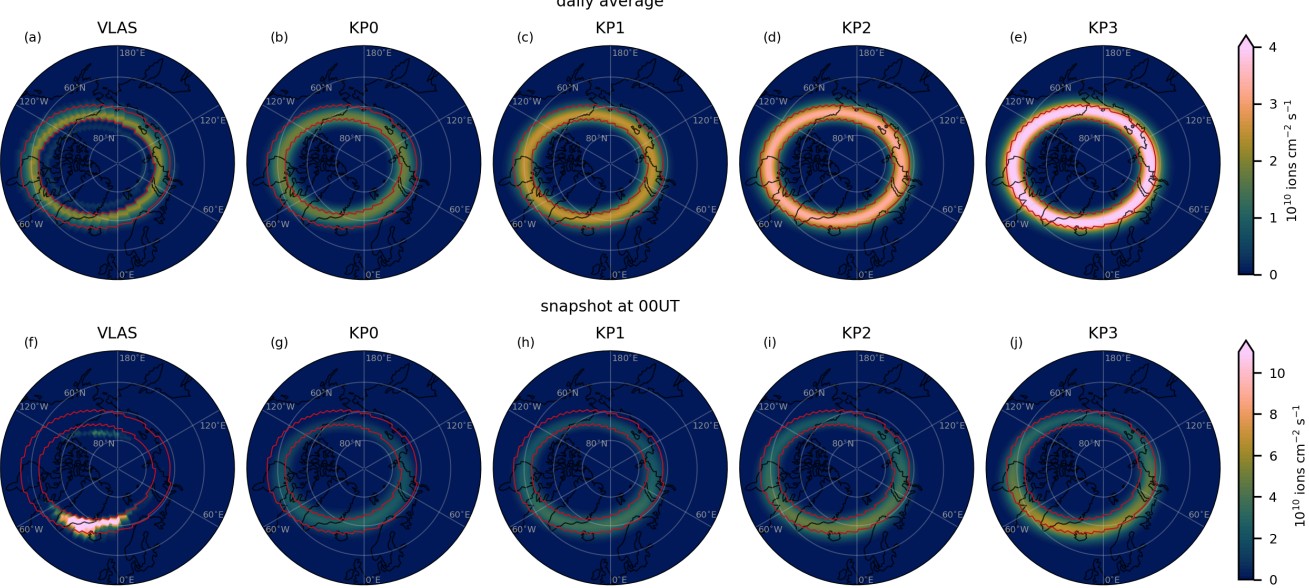

Altitude-integrated auroral ionisation rates, NH, 1 Jan 2006

**Figure 4.** Vertically integrated auroral ionisation rates on geographic coordinates for the northern hemisphere (geographic latitude $> 50°$N). 1 Jan 2006, (a–e) daily average and (f–j) snapshot at 00 UT. (a, f) VLAS, (b, g) KP0, (c, h) KP1, (d, i) KP2, (e, j) KP3. For comparisons, the red lines indicate the region where the VLAS integrated daily average auroral ionisation exceeds $0.5 \times 10^{10}\,\mathrm{ions\,cm^{-2}\,s^{-1}}$.

auroral ionisation. Especially on the poleward edge of the auroral oval the eVlasiator-derived forcing seems to have a sharp drop-off, which can also be seen in the latitudinal–altitudinal extent of the auroral ionisation shown in Fig. 5b. The sharpness is even more apparent in the instantaneous ionisation rates from eVlasiator, Figs. 4f, 6b. The eVlasiator nightside sharp poleward

edge is in line with previous studies (e.g. Newell et al., 1996), while the $Kp$ parametrisation by Roble and Ridley (1987) on the other hand includes so-called "polar rain" electron precipitation, which extends as a uniform distribution over the geomagnetic pole, and thus "softens" the poleward edge of the auroral electron forcing. The polar rain can be seen for the KP2 run in the cross-sections in Figs. 5c, 6c extending towards the pole from the auroral oval region.

The eVlasiator forcing also shows a slight secondary peak in the daily average ionisation on the poleward side in Fig. 4a, and

more clearly in the cross-section in Fig. 5b. Comparing the daily averages to the instantaneous eVlasiator-derived ionisation, shown in Fig. 4f, we see that the two-peak structure in the daily average ionisation results from the dayside and nighside auroral ionisations. The secondary peak comes from the dayside ionisation forcing being located closer to the (geomagnetic) pole than the nightside ionisation.

The clear separation into the nightside and dayside ionisation peaks is also a clear difference between the eVlasiator auroral

electron forcing and the $Kp$ parametrisation. While similar to the eVlasiator ionisation in that the nightside has higher ionisation rates than the dayside, the parameterisation shows a continuum between day and night. The eVlasiator on the other hand

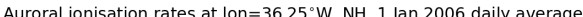

**Figure 5.** The daily average auroral electron ionisation rates along geographic longitude 36.25°W on 1 Jan 2006. (a) Maximum ionisation rates from auroral electron forcing at each altitude along the longitude 36.25°W for the VLAS and KP0–KP5 simulations. (b–c) Geographic latitude–altitude extent of the auroral electron forcing, as LOG10 of the auroral ionisation rates for the (b) VLAS and (c) KP2 simulations.

has two clear peaks along the auroral oval, dayside much weaker than the nightside, with little ionisation in the dusk and dawn sectors. This structure of day/night peaks matches with how the DMSP scaling of the eVlasiator fluxes was applied.

There is also a difference on the equatorward side of auroral oval between the eVlasiator-derived ionisation and the $Kp$
parametrisation, as the eVlasiator auroral electron forcing stops. This difference in extent results from the limitation in the latitudinal coverage made possible by the used eVlasiator run, as the cutoff latitude of 63° MLAT corresponds to the mapping of the innermost considered locations in the magnetospheric domain (4.8 $R_E$; see Appendix A). In future runs with an inner boundary closer to the Earth's surface, this sharp cutoff near the auroral oval's equatorward boundary could be avoided.

The eVlasiator auroral ionisation forcing reaches deeper into the atmosphere down to around 0.01 hPa ($\sim$80 km), while the
$Kp$ parameterisation does not extend below $5\times10^{-4}$ hPa ($\sim$95 km). Though the eVlasiator ionisation rates are negligible at the 0.01 hPa level, the tapering off of the aurora towards lower altitudes is more gradual compared to the $Kp$ parametrisation. The eVlasiator forcing also peaks at a slightly lower altitude compared to the $Kp$ parametrisation. This can be seen in Fig. 5a, which

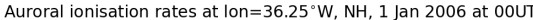

Auroral ionisation rates at lon=36.25°W, NH, 1 Jan 2006 at 00UT

**Figure 6.** Same as Fig. 5, but for instantaneous ionisation rates on 1 Jan 2006 at 00 UT.

shows the maximum ionisation rates at each altitude for the VLAS and KP0–KP5 simulations. We can also see that towards the model top eVlasiator on average produces less ionisation than even the KP0 case, but in the instantaneous ionisation rates

VLAS has more ionisation than KP3 throughout the vertical extent of the $Kp$ parametrisation (Fig. 6a). The nighttime peak of the eVlasiator-derived ionisation is much stronger than the $Kp$ parametrisation, with more than an order of magnitude difference even to KP5 at 95 km altitude.

## 3.2  Atmospheric impact

Figure 7 shows the altitude-integrated $NO_x$ response averaged over the polar region (geographic latitude $> 60°$) in the auroral

runs relative to the REF run with no auroral electron forcing. The polar averages have been calculated by weighting with the cosine of the geographic latitude to account for the increasing gridpoint density towards the poles. For both SH and NH we clearly see the $NO_x$ impact during the winter season for all the auroral forcing scenarios. The effect is naturally strongest in the thermosphere, where the auroral electrons have a direct impact, with the eVlasiator auroral precipitation causing a $NO_x$ increase of up to 215 % (from $1.62 \times 10^{14}$ molecules cm$^{-2}$ in REF to $5.13 \times 10^{14}$ molecules cm$^{-2}$ in VLAS) in the SH lower

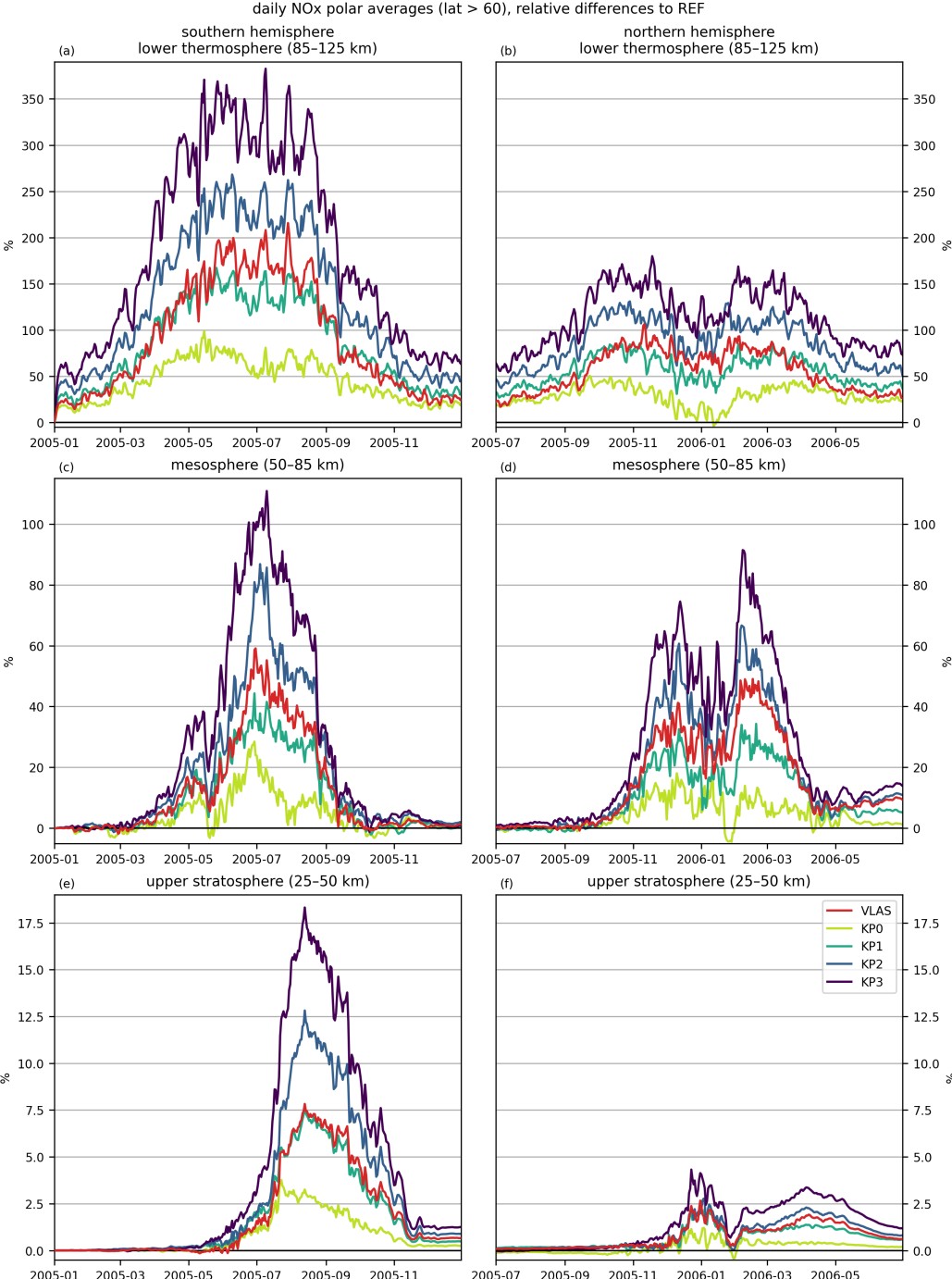

**Figure 7.** Auroral impact on $NO_x$ concentrations: Polar averages (geographic latitude $> 60°$) of the integrated (a–b) lower thermosphere, (c–d) mesosphere, and (e–f) upper stratosphere for the (a, c, e) SH and (b, d, f) NH winters. Relative difference of the auroral precipitation simulation runs (colours) compared to the REF simulation. Lower thermosphere integrated from $3 \times 10^{-3}$ hPa to $1 \times 10^{-5}$ hPa ($\sim$85–125 km), mesosphere from $1$ hPa to $3 \times 10^{-3}$ hPa ($\sim$50–85 km), and upper stratosphere from $30$ hPa to $1$ hPa ($\sim$25–50 km).

thermosphere ($\sim$85–125 km). The descent of the produced $NO_x$ can also clearly be seen in the mesosphere ($\sim$50–85 km) and upper stratosphere ($\sim$25–50 km), where $NO_x$ is increased by up to 59 % (from $3.06 \times 10^{14}$ molecules cm$^{-2}$ to $4.87 \times 10^{14}$ molecules cm$^{-2}$) and 7.8 % (from $1.61 \times 10^{15}$ molecules cm$^{-2}$ to $1.74 \times 10^{15}$ molecules cm$^{-2}$), respectively. The corresponding VLAS peak $NO_x$ impacts in the NH are 106 % (from $2.30 \times 10^{14}$ molecules cm$^{-2}$ to $4.74 \times 10^{14}$ molecules cm$^{-2}$), 49 % (from $2.65 \times 10^{14}$ molecules cm$^{-2}$ to $3.95 \times 10^{14}$ molecules cm$^{-2}$), and 2.7 % (from $1.46 \times 10^{15}$ molecules cm$^{-2}$ to $1.50 \times 10^{15}$ molecules cm$^{-2}$) for lower thermosphere, mesosphere, and upper stratosphere, respectively. In the SH the descent of the $NO_x$ produced by auroral electrons is also evident in the lag in the peak occurrence times, as the strongest SH thermospheric, mesospheric, and stratospheric $NO_x$ impacts occur in June, July, and August, respectively. The impact also grows weaker during the descent, since the background levels of $NO_x$ are generally higher at the lower altitudes, and not all the produced $NO_x$ descends.

Comparing the $Kp$-driven auroral precipitation runs, there is a clear scaling effect in Fig. 7: the $NO_x$ impact grows stronger with increasing $Kp$. Even in the KP0 case the SH lower thermospheric $NO_x$ is increased by 98% (from $1.35 \times 10^{14}$ molecules cm$^{-2}$ to $2.68 \times 10^{14}$ molecules cm$^{-2}$), with KP3 reaching an increase of over 380% (from $1.68 \times 10^{14}$ molecules cm$^{-2}$ to $8.12 \times 10^{14}$ molecules cm$^{-2}$) in the SH. The eVlasiator auroral forcing run (VLAS) corresponds to the KP1 and KP2 simulations in terms of the $NO_x$ impact, often coming closer to the KP1 scenario. This is despite the weaker daily average ionisation rates seen in Fig. 4 from the DMSP-scaled eVlasiator electron fluxes. It seems that the strong nighttime peak ionisation in the eVlasiator run compensates for the lack of continuous auroral forcing seen in the $Kp$ parametrisation. The reverse may also be true: the $Kp$ parametrisation may be compensating for a lack of high enough ionisation rates by applying relatively high ionisation throughout the day.

The strongest impacts are consistently seen in the southern hemisphere. The differences between the hemispheres can be explained by the instability of the polar vortex in the NH, as well as the sudden stratospheric warming (SSW) that occurred in mid-January 2006 (Manney et al., 2008; Butler et al., 2015). This is likely the cause of the double peaks in the $NO_x$ impact in the NH, since the anomalous dynamical conditions due to SSW result in increased $NO_x$ levels in all the WACCM simulations, including REF, so the relative differences decrease during the SSW. Later in the winter strong downward transport resumed and caused a sharp increase of $NO_x$ in the mesosphere (Randall et al., 2009), which is also seen in our simulations.

The difference in the descent of $NO_x$ between the two hemispheres can also be seen in the $NO_x$ profiles shown in Fig. 8. The profiles are averaged over the polar region (geographic latitude $> 60°$) with cosine weighting, as in Fig. 7, and additionally over the winter months for each hemisphere. In the NH there is very little difference between the auroral runs and the REF simulation at stratospheric altitudes, whereas in the SH the runs are distinguishable from each other down to about 7 hPa level. This is due to the more efficient downward transport of $NO_x$ within the polar vortex in the SH than NH. As in Fig. 7, the VLAS $NO_x$ profiles most closely correspond to the KP1 and KP2 scenarios. We also see the scaling of the $Kp$ parametrisation as the impact get progressively stronger from KP0 to KP3.

For more details on the spatial distribution of the auroral precipitation $NO_x$ impact, Fig. 9 shows the REF wintertime averages in both hemispheres, and the increase in the VLAS simulation relative to REF. The REF number densities show a difference in the vertical distribution of $NO_x$ between the hemispheres, also visible in the profiles in Fig. 8. In the north, $NO_x$

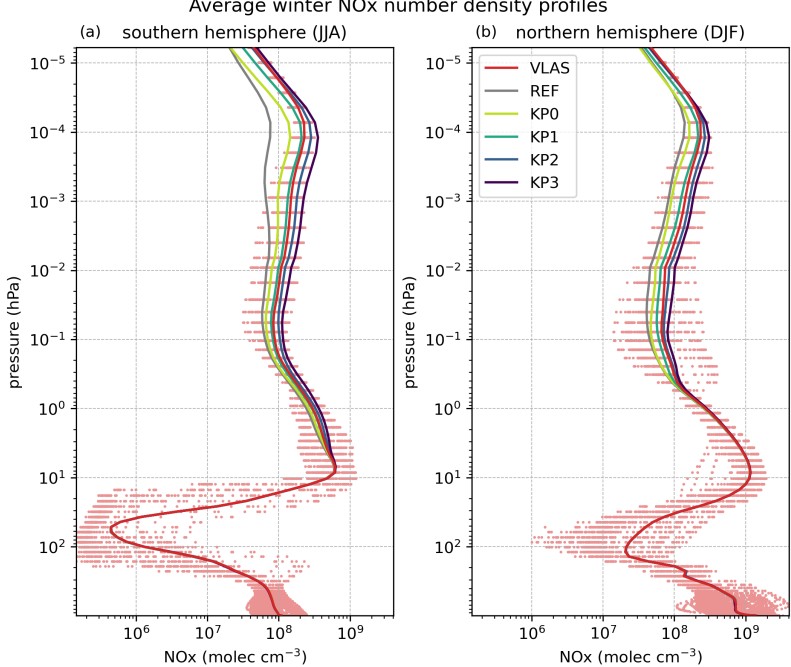

**Figure 8.** Average $NO_x$ profiles (solid lines) during wintertime for the REF, VLAS, and KP0–KP3 simulations in the (a) SH (Jun–Aug 2005) and (b) NH (Dec 2005–Feb 2006) polar regions (geographic latitude $>60°$). The dots represent daily $NO_x$ number densities in the VLAS simulation.

has a stronger polar peak at thermospheric altitudes than SH, whereas at mesospheric altitudes on SH has more $NO_x$ than NH. The NH themospheric peak is visible in the REF $NO_x$ profile (Fig. 8b), while at mesospheric altitudes there seems to be a drop in the REF $NO_x$ concentration. The mesospheric $NO_x$ drop does not seem to be present in the SH (Fig. 8a), leading to the difference in $NO_x$ levels seen in Fig. 9b, e. The occurrence of a solar proton event in mid-May 2005 likely contributes to the SH $NO_x$ in the mesosphere, as the proton precipitation penetrates to mesospheric altitudes. The increased $NO_x$ production

and the longer chemical lifetime in the winter pole allows $NO_x$ to accumulate in the SH mesosphere. In the NH the SSW may also be leading to disruption in the vertical transport of $NO_x$ so that a greater proportion of produced $NO_x$ stays in the thermosphere rather than being transported downward.

The spatial distribution of $NO_x$ in the REF simulation also shows the difference in the stability of the polar vortices. In the SH mesosphere and upper stratosphere $NO_x$ is rather symmetrically distributed around the pole, due to the stable polar

vortex during the winter. In the NH, on the other hand, the less stable polar vortex results in both the mesospheric and upper stratospheric $NO_x$ concentrations being distibuted assymmetrically and off-pole. The mesospheric $NO_x$ peak is also slightly shifted compared to the stratospheric $NO_x$ trough, indicating vertical shifts in the polar vortex. The SSW that occurred during the 2005–2006 NH winter likely adds to the instability of the NH polar vortex.

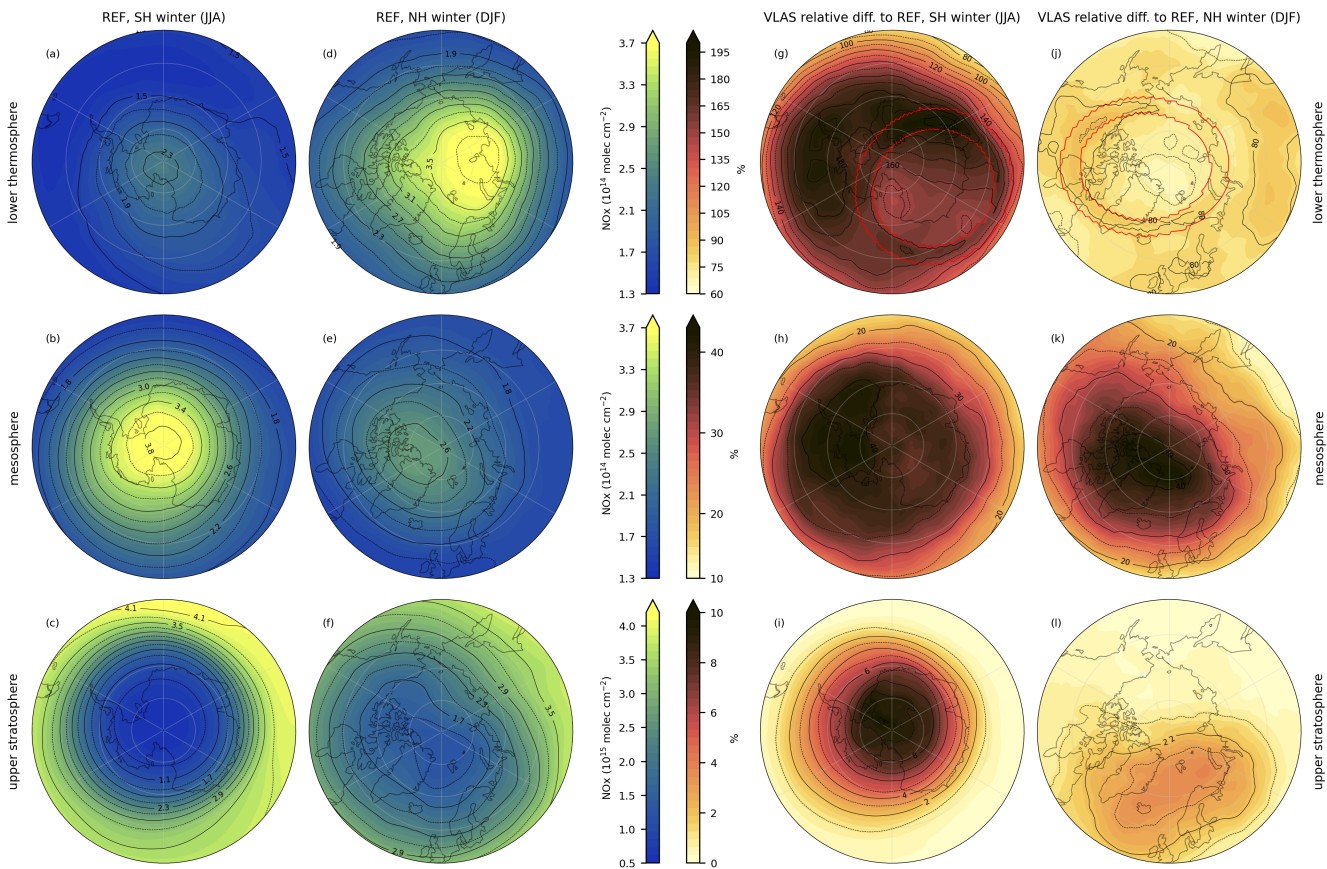

**Figure 9.** (a–f) Altitude integrated REF $NO_x$ number densities and (g–l) the relative $NO_x$ impact of the VLAS simulation compared to REF, for (a–c, g–i) SH and (d–f, j-l) NH polar regions (geographic latitude $> 50°$). Both the the concentrations and relative differences are averaged over the winter seasons (Jun–Aug 2005 for SH, Dec 2005–Feb 2006 for NH). The red lines show the auroral oval by indicating the region where the VLAS integrated daily average auroral ionisation exceeds $0.5 \times 10^{10} \, \mathrm{ions \, cm^{-2} \, s^{-1}}$ on (g) 1 Jul 2005 and (j) 1 Jan 2006, as in Fig. 4. The atmospheric layers – (a, d, g, j) lower thermosphere, (b, e, h, k) mesosphere, and (c, f, i, l) upper stratosphere – correspond to Fig. 7.

In the VLAS run, $NO_x$ is clearly increased throughout the thermosphere and mesosphere, and not just in the auroral oval latitudes, as $NO_x$ is transported from the production region. In the upper stratosphere, the impact is confined inside the polar vortex latitudes with little impact outside it. The SH lower thermosphere does show the effect of the electron precipitation both inside and outside the auroral oval, with the strongest $NO_x$ responses reaching over 200 %, corresponding to Fig. 7a. The NH thermospheric $NO_x$ response is much weaker than the SH, and the auroral oval pattern can only be distinguished on the North American longitude sector. The weaker response and its longitudinal distribution can overall be explained by the REF $NO_x$ levels being higher in the NH than the SH, and by the location of the NH thermospheric $NO_x$ peak over the Eurasian longitude sector.

In the mesosphere, the relative auroral precipitation impact is stronger in the NH than the SH, again explained by the difference in the REF background levels. This corresponds well to the mesospheric VLAS impacts in Fig. 7c–d. Stratospheric impacts show again the differences in the REF $NO_x$ number densities between the hemispheres. The SH shows clearly the impact centred around the pole, since the $NO_x$ produced by the auroral precipitation descends within the polar vortex from thermospheric altitudes to the startosphere. In the NH the VLAS auroral electron impact in the stratosphere has a more irregular form, and it is located off-pole, showing the less stable NH polar vortex.

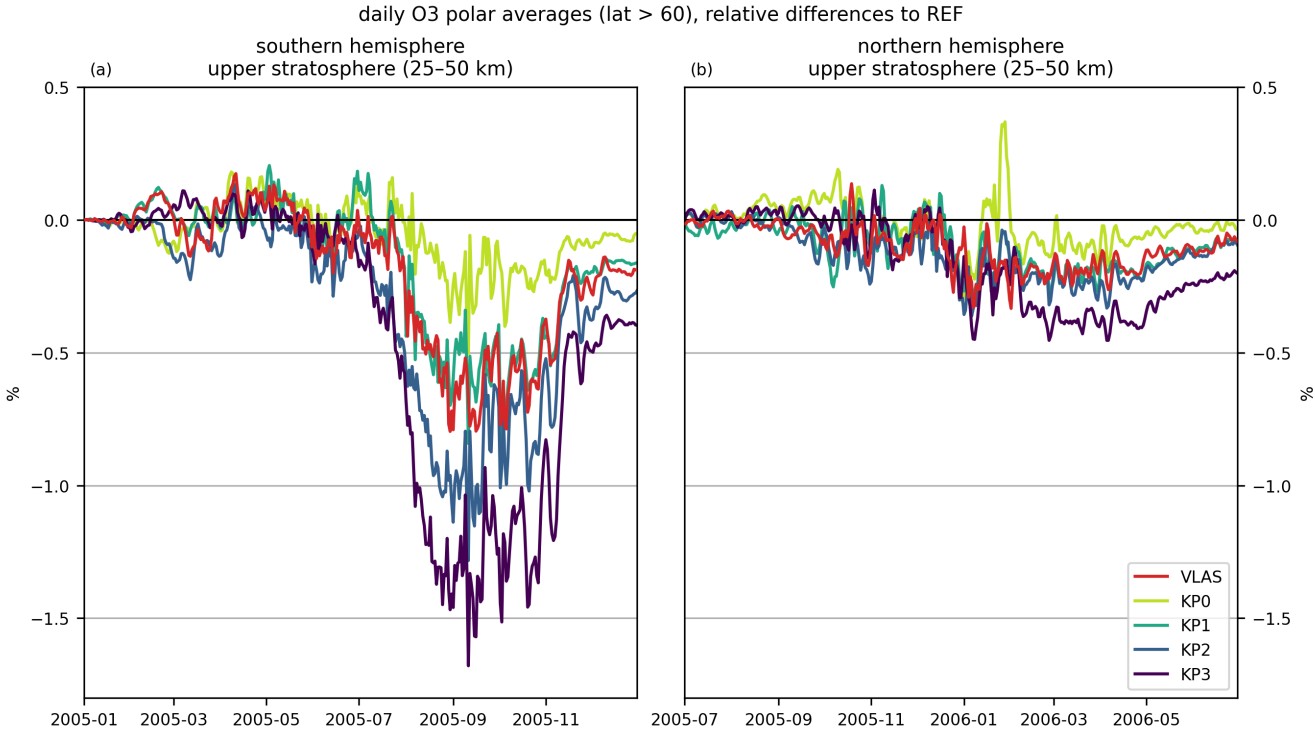

**Figure 10.** Upper stratospheric ozone response to auroral electron forcing: daily relative difference in vertically integrated ozone number densities from the auroral precipitation simulation runs (colours) compared to the REF run with no auroral precipitation. Polar averages (geographic latitude $> 60°$) for (a) SH and (b) NH winters.

The upper stratospheric ozone responses to the auroral electron forcing scenarios are shown in Fig. 10. The $O_3$ impact is much weaker than $NO_x$, with a peak decrease of 0.80 % (from $1.829 \times 10^{18}$ molecules $cm^{-2}$ to $1.814 \times 10^{18}$ molecules $cm^{-2}$) in SH upper stratospheric $O_3$ in the VLAS simulation. Comparing the KP3 SH results from Figs. 7 and 10, we see that a $NO_x$ increase of over 380 % in the lower thermosphere leads to an increase of around 18 % $NO_x$ in the upper stratosphere, corresponding to a reduction in upper stratospheric $O_3$ by only 1.68 % (from $2.048 \times 10^{18}$ molecules $cm^{-2}$ to $2.014 \times 10^{18}$ molecules $cm^{-2}$). As with $NO_x$, the ozone impact is stronger in the SH than in the NH, where the VLAS upper stratospheric ozone decrease is around 0.33 % (from $2.803 \times 10^{18}$ molecules $cm^{-2}$ to $2.794 \times 10^{18}$ molecules $cm^{-2}$). Even in the KP3 case the NH impact is only 0.45 % (from $2.747 \times 10^{18}$ molecules $cm^{-2}$ to $2.735 \times 10^{18}$ molecules $cm^{-2}$) when compared to the reference run.

## 4  Discussion

Our results demonstrate a successful one-way interfacing of magnetospheric and atmospheric simulations. We are able to produce realistic auroral electron precipitation fluxes from eVlasiator, and they have been applied as auroral electron forcing in the WACCM model. The simulated electron fluxes produce atmospheric impacts comparable to WACCM's current auroral electron parameterisation, but with enhanced information on energy and spatial distribution. Thus this work presents the potential for future studies on the effects of the solar wind on the atmosphere, e.g. for the study of the atmospheric impacts of magnetospheric substorms. Eventually, atmospheric forcing could be driven directly by solar wind parameters instead of proxy-based parameterisations built on limited magnetospheric electron flux data. Since solar wind parameters can be observed earlier than e.g. the geomagnetic activity determining the $Kp$ index, this could lead to improved, near-real-time predictions of the atmospheric response in the future. First steps towards this include the production of time-dependent auroral electron precipitation forcing from the magnetospheric simulations, on an extended temporal scale more useful to long-term atmospheric simulations.

Our results show a clear difference in the structure of the auroral electron forcing between the $Kp$ parameterisation and eVlasiator. While eVlasiator produces a high nighttime peak in ionisation coupled with a much weaker daytime peak, the $Kp$ parameterisation applies ionisation forcing throughout the day, with much less diurnal variability. This leads to a difference in the daily average ionisation rates as well. Comparing the VLAS, KP2, and KP3 cases, the parametrisation produces on average higher ionisation than the scaled eVlasiator auroral electron fluxes, but the eVlasiator nighttime peak integrated ionisation is greater than the KP3 peak by a factor of approximately 2.5. This is despite scaling the eVlasiator electron fluxes with the DMSP observations during $Kp$ index values of 2–3. On average the $Kp$ parameterisation may therefore be overestimating the auroral electron forcing, but the lack of a strong nighttime peak seems to at least partially mitigate the overestimation. The new method presented here provides a unique approach to auroral forcing, independent of electron flux measurements, but further studies are needed to ascertain the correct level of auroral electron precipitation, as well as resulting $NO_x$ impacts. Satellite observations of $NO_x$ species could be used to study the accurate levels of $NO_x$ production from auroral electrons to evaluate the model results. This study also uses single WACCM-D runs. While we use the specified dynamics, the auroral electron precipitation occurs well within the free-running altitude range (above 50 km) of WACCM. Ensemble simulations would provide more

robust model results on the magnitude of the impact of auroral electrons. eVlasiator simulations of the auroral electron fluxes with conditions corresponding to higher $Kp$ indices should be carried out as well. In addition, electrons at energies beyond the auroral range ($> 30$keV) should also be considered, e.g. through the inclusion of reconnection and radiation belt processes in future versions of Vlasiator. This could aid in bridging the possible gap between auroral and medium-energy electrons.

Limitations of the magnetospheric models should also be considered. As pointed out in Sect. 2.3.1, eVlasiator does not model all sources of precipitating auroral electrons, and therefore the obtained precipitating fluxes might differ from reality. We have mitigated the effect of this possible discrepancy in this study by using the DMSP observations to scale the electron fluxes, although the scaling can only increase eVlasiator fluxes at energies for which the values are non-zero. For this reason, the high-energy cutoff associated with the sparse description of phase-space density in eVlasiator remains even after the scaling,

which translates to a limit altitude below which the eVlasiator fluxes cannot produce ionisation in the atmosphere. Since we have included the CMIP6 recommended medium-energy electron forcing (energies $> 30$keV) in our atmospheric simulations, we excluded the eVlasiator forcing at corresponding energies. Future work should consider the combination of the different electron precipitation sources with possibly overlapping energy spectrums in detail.

The latitudinal extent of the eVlasiator-derived auroral precipitation is also limited compared to the $Kp$ parameterisation. On

the equatorward edge of the auroral oval this arises from the distance of the eVlasiator run's inner boundary from the surface of the Earth. On the poleward side the difference is partially explained by the inclusion of polar rain in the $Kp$ parameterisation. We have not considered these differences and limitations in the interpretation of the atmospheric impacts of the precipitation. Auroral ionisation rates for future eVlasiator simulations with a less sharp cutoff on the equatorward side of the auroral oval will therefore likely provide an enhancement in the $NO_x$ response, and, as seen in Fig. 9, the $NO_x$ impact is not limited to the

auroral oval region.

## 5    Conclusions

In this study we have demonstrated for the first time a novel approach to investigating the role of auroral electron precipitation in the MLTI. We used eVlasiator to simulate electron precipitation fluxes at auroral energies ($50$ eV–$50$ keV) that were scaled using satellite observations to account for deficiencies in the magnetospheric model. Ionisation rates derived from the electron

fluxes were then used as input in WACCM-D in order to analyse the atmospheric $NO_x$ and ozone impacts of the auroral electron precipitation. We found the strongest response in the SH polar lower thermosphere, where the eVlasiator-derived auroral precipitation increased $NO_x$ concentrations up to 215 % (from $1.62 \times 10^{14}$ molecules cm$^{-2}$ to $5.13 \times 10^{14}$ molecules cm$^{-2}$). In the mesosphere there was an increase of 59 % (from $3.06 \times 10^{14}$ molecules cm$^{-2}$ to $4.87 \times 10^{14}$ molecules cm$^{-2}$) in $NO_x$ in SH, with NH also reaching an increase of 49 % (from $2.65 \times 10^{14}$ molecules cm$^{-2}$ to $3.95 \times 10^{14}$ molecules cm$^{-2}$). The

auroral precipitation response can also be seen in the upper stratosphere, where we see a $NO_x$ increase of around 7.8 % (from $1.61 \times 10^{15}$ molecules cm$^{-2}$ to $1.74 \times 10^{15}$ molecules cm$^{-2}$), which corresponds to a peak decrease of 0.80 % (from $1.829 \times 10^{18}$ molecules cm$^{-2}$ to $1.814 \times 10^{18}$ molecules cm$^{-2}$) in upper stratospheric ozone.

As comparison to the eVlasiator-derived auroral precipitation we used WACCM's parameterisation of auroral electron forcing, which is driven by the $Kp$ index based on the auroral model by Roble and Ridley (1987). Overall, the electron precipitation from eVlasiator is similar to the parameterised auroral electron forcing in location and impact, although there are clear differences in the structure of the auroral forcing. While eVlasiator produces a strong nighttime peak in ionisation rates, the parameterisation on average has more ionisation. The latitudinal extent of the eVlasiator auroral electron precipitation is partially limited, and the average ionisation rates are somewhat weaker than the parameterisation, even with the satellite observation-based scaling of the electron fluxes. On the other hand eVlasiator provides more detailed energy and spatial distributions of the auroral electron precipitation, with a clear nighttime peak, and the ionisation forcing reaches deeper in to atmosphere, down to 80 km compared to around 95 km in the parameterisation.

As a next step, in order to validate the accuracy of the model results, specific simulations could be carried out for comparisons with satellite observations. For this, time-dependent auroral electron precipitation data from eVlasiator would be needed in order to model the variability of auroral electron impact in the atmosphere. For example, impacts should be studied during the different phases of substorms. For the future, this work paves the way for a more complete description of auroral electron forcing in atmospheric simulations and, eventually, for the detailed study of solar wind – atmosphere interaction.

*Code and data availability.* The Vlasiator code is open-source under GPL-2, indexed through Zenodo (Pfau-Kempf et al., 2024), and available through GitHub. The eVlasiator release used for this study is similarly available at Pfau-Kempf et al. (2022). The Vlasiator simulation data used for this study consists of several terabytes of data, and is thus not made available online, but the authors accept data requests. The reduced output of the eVlasiator simulation consisting of precipitating electron differential number flux data is available at Finnish Meteorological Institute Research Data repository METIS (Grandin, 2024).

The DMSP/SSJ precipitating particle fluxes are openly available and were retrieved from http://cedar.openmadrigal.org/.

WACCM-D simulation data analysed in this paper are available at Finnish Meteorological Institute Research Data repository METIS (Häkkilä and Szelag, 2024a, b, c). The new WACCM IPR code enabling MLT-dependent ionisation input is indexed via Zenodo (Häkkilä, 2024), and publicly available on GitHub at https://github.com/hakkila/waccm_iprmlt.

## Appendix A: Detailed description of the mapping between the ionosphere grid and the eVlasiator simulation domain

To produce an auroral-electron forcing dataset for WACCM-D, we need to map the fluxes obtained with eVlasiator to ionospheric altitudes. The procedure detailed below is illustrated in Fig. A1.

Magnetic field lines (in magenta) are followed between the ionosphere (at an altitude of 110 km; points $A_i$) and the inner boundary of the simulation domain from start points placed every $1°$ in MLAT and 0.5 h in MLT. Since the magnetic field inside the inner boundary only consists of the Earth dipole and has no perturbed field term, we construct a more realistic mapping by superposing two magnetic field components. The internal contributions to the geomagnetic field are described by a simple point dipole to match the geomagnetic field description used in Vlasiator. The Tsyganenko 2001 (T01; Tsyganenko, 2002a, b) model is used to describe the external field contributions, with the solar wind conditions of the Vlasiator run (see Sect. 2.1), at

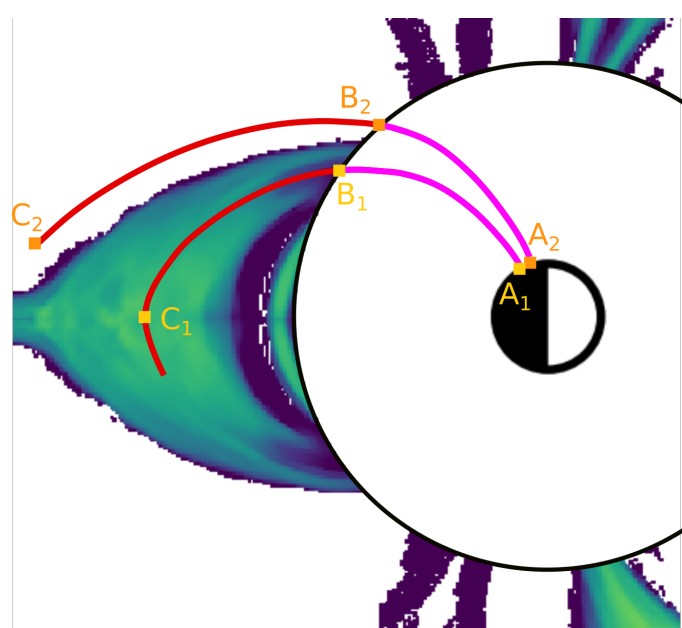

**Figure A1.** Illustration of the mapping of eVlasiator precipitating electron fluxes to ionospheric altitudes. The view is a slice of the eVlasiator domain in the noon–midnight meridional plane, with the Sun located to the right of the figure. Points $A_1$ and $A_2$ are located on the ionospheric grid (Cartesian in MLAT–MLT) at 110 km altitude. Points $B_1$ and $B_2$ are located next to the eVlasiator domain's inner boundary at 4.8 $R_E$. The magenta lines indicate the superposition of a non-tilted dipole field with the T01 model. The red lines follow the magnetic field within the eVlasiator domain. Point $C_1$ is located in the equatorial plane, and point $C_2$ is located at a distance of 7.5 $R_E$ from point $B_2$ along the magnetic field line.

a date when the geomagnetic dipole was almost perpendicular to the ecliptic plane (11 March 2020, 21:40 UT), and assuming a Dst value of $-30$ nT. The Python versions of T01 and Earth dipole field implemented in the geopack library (Tian, 2023) were used for this mapping of atmospheric altitudes to 4.8 $R_E$, i.e. just beyond the inner boundary (points $B_i$). From each grid point, we follow the geomagnetic field obtained by combining the untilted dipole model for internal contributions with the T01 model for external contributions. Up to this step, the procedure is the same as described in more detail in Grandin et al. (2023).

Within the 1.4 s of the eVlasiator run, electrons scattered into the bounce loss cone in the plasma sheet do not have time to reach the inner boundary of the simulation domain. To account for this, we extend the mapping of the MLT–MLAT grid further into the eVlasiator simulation domain so as to reach the magnetospheric regions where precipitating electrons originate. To be consistent, we use the magnetic field from Vlasiator (in red) to extend the mapping outwards for another 7.5 $R_E$, or until reaching the equatorial plane. The value of 7.5 $R_E$ was empirically determined; it ensures that this distance is sufficient to reach

the transition region for all the closed field lines on the nightside without extending unnecessarily far down the magnetotail or in the cusp for the open field lines.

        Finally, along each field line thus obtained and for each electron energy bin, the maximum value of the precipitation differential flux between the inner boundary (points $B_1$, $B_2$) and either the point where the magnetic field tracing was stopped

(point $C_2$) or the equatorial plane (point $C_1$) is retained. This ensures that precipitating electrons which may not have had time
to reach the inner boundary by the end of the electron simulation are taken into account, and gives a conservative high value
for the differential number flux.

## Appendix B: Detailed description of the scaling of eVlasiator electron fluxes with DMSP/SSJ observations

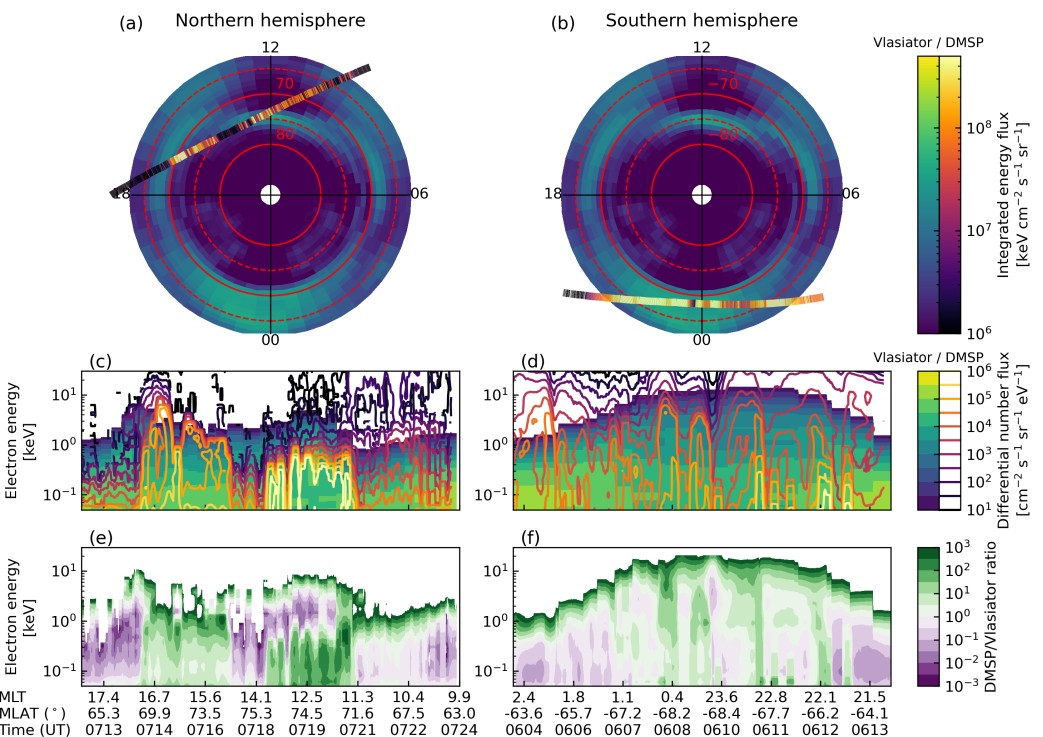

**Figure B1.** Comparison of eVlasiator and DMSP/SSJ observations during the 1 Aug 2011 overpasses. (a–b) Integrated energy flux of precipitating electrons obtained with eVlasiator (background) on top of which corresponding measurements by DMSP/SSJ (contours) are overlaid along the spacecraft's orbits. (c–d) Differential number fluxes along the orbit for eVlasiator (background colour) and DMSP/SSJ (contours). (e–f) Ratio between DMSP/SSJ and eVlasiator differential number fluxes along the orbits.

### B1   Comparison of eVlasiator fluxes along the DMSP orbits

Figures B1 and B2 show the comparison of eVlasiator precipitating electron fluxes with DMSP/SSJ measurements during
the two events with similar driving conditions as in the Vlasiator run (8 Aug 2011 and 10 Oct 2015). The top panels show
the global view of eVlasiator integrated energy fluxes in both hemispheres, on top of which DMSP/SSJ integrated energy
fluxes along the spacecraft's orbits are overlaid. The middle panels enable the comparison of differential number fluxes of

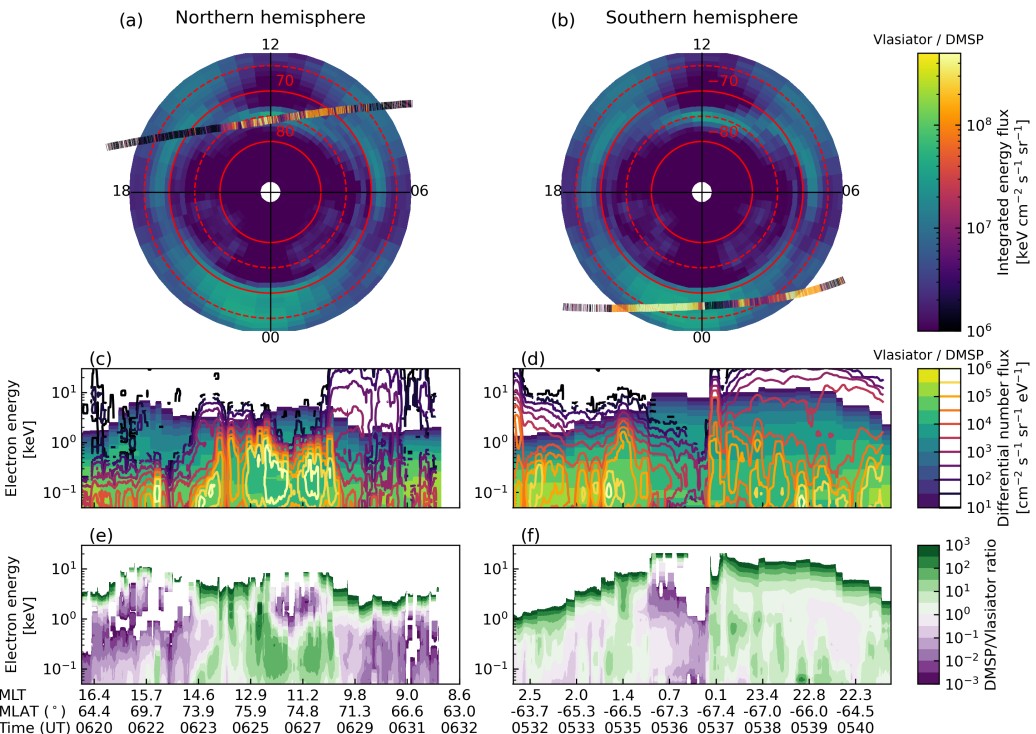

**Figure B2.** Comparison of eVlasiator and DMSP/SSJ observations during the 10 Oct 2015 overpasses. Same format as in Fig. B1.

precipitating electrons. One can see in particular that overall eVlasiator spectra have a cutoff at high energies, indicating that the high-energy component is often missing compared to DMSP/SSJ observations. This is due to the sparsity threshold used in eVlasiator simulations, which discards velocity cells within which the phase-space density is below the threshold to keep the simulation computationally feasible. Since the phase-space density decreases near the edges of the velocity distribution, applying the sparsity threshold creates a sharp drop at those edges, which translates into a cutoff at high energies in the precipitating flux.

The bottom panels show the ratio between the differential number fluxes measured by DMSP/SSJ and obtained with eVlasiator are displayed, along the satellites' orbits. Green regions correspond to eVlasiator underestimating the electron precipitation, whereas purple regions correspond to eVlasiator overestimating it, when considering a given event and location along the orbit. The ratios typically range between 0.001 and 1000, highlighting the need for a scaling of the eVlasiator fluxes so that they might be more realistic. It is clear from those bottom panels that the correction to be applied to eVlasiator fluxes must be different on the dayside and on the nightside, and that it must be energy-dependent. Below we detail how the corrected eVlasiator fluxes were determined and we justify the choices made in developing the method.

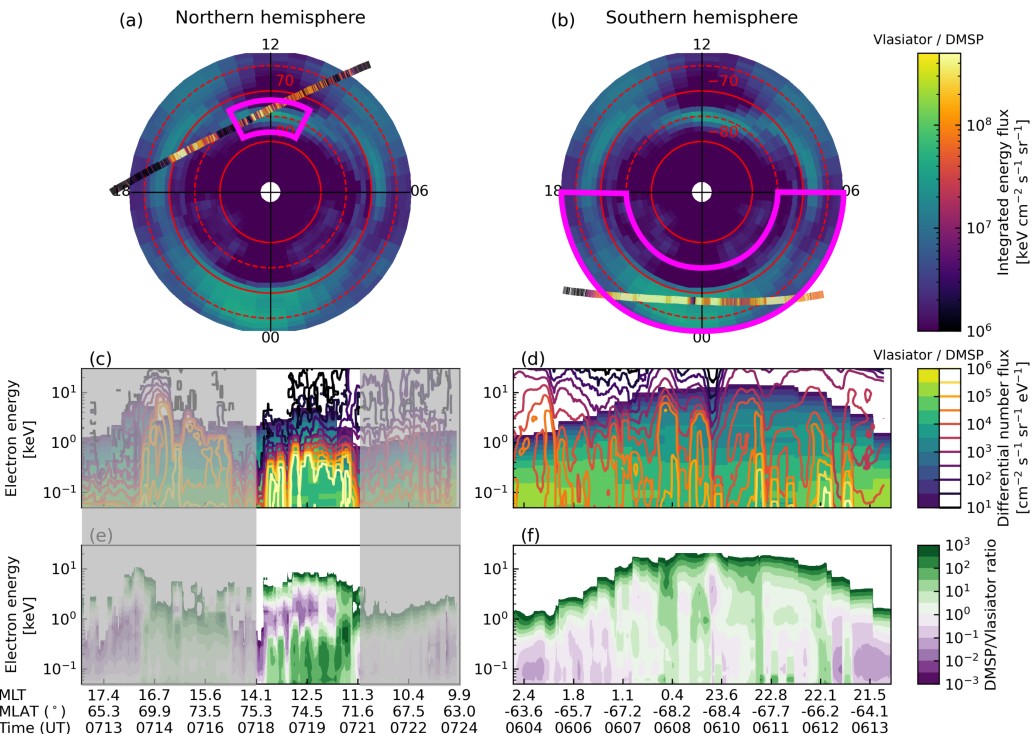

**Figure B3.** Same as Fig. B1 with regions of interest (cusp and nightside oval) indicated in magenta in panels a–b. The grey shading in panels c and e indicates the masking used for the dayside overpass, to keep only cusp measurements.

## B2 Selection of regions of interest to be corrected

In order to avoid increasing the precipitating electron fluxes outside of the auroral oval (e.g. in the polar cap or in the flanks, where eVlasiator fluxes might be contaminated by boundary effects), we restrict the correction to the cusp and nightside oval regions, as indicated with magenta contours in Fig. B3a–b. The same regions of interest are used for both events. Besides, extremely low eVlasiator flux values ($< 0.01 \, \mathrm{el \, cm^{-2} \, s^{-1} \, sr^{-1} \, eV^{-1}}$) are masked too. The masked data are shown in grey in Fig. B3c,e.

## B3 Percentile fitting for DMSP/eVlasiator flux ratios

Since we want to obtain correction coefficients for the eVlasiator fluxes as a function of electron energy, we need to find a suitable metric to derive such coefficients based on the ratios between DMSP and eVlasiator fluxes along the DMSP orbits. A quick inspection of Fig. B3e–f reveals that using the mean value (along the DMSP orbits) of the ratio at a given energy would not provide a robust estimate of the needed correction, since for instance at $1 \, \mathrm{keV}$ on the nightside values range from $10^3$ (start of the orbit) to $< 1$ (middle of the orbit), which would result in a mean value skewed to the high values and not

necessarily representative of the needed correction coefficient at $1\,\mathrm{keV}$. This is because the eVlasiator fluxes drop off quickly at the high-energy end of their spectra, due to the sparse velocity space description (Palmroth et al., 2018). Therefore, instead of using the mean, we use a percentile value as the metric to determine the energy-dependent correction coefficients (one set for cusp fluxes, one set for nightside fluxes).

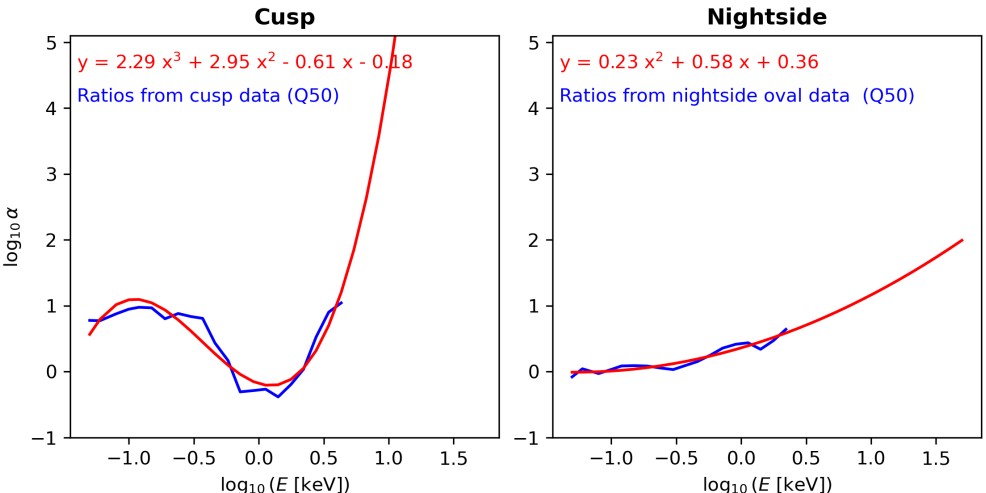

**Figure B4.** Energy-dependent correction coefficients obtained for the cusp (left) and nightside (right) eVlasiator fluxes obtained by taking the median (Q50) values of the DMSP/eVlasiator flux ratios. The red lines indicate polynomial fits of the data-based curves in blue.

Figure B4 shows results obtained when considering median (50th percentile) values. We see that the median values of the ratios (blue lines) can be fitted with a third-order polynomial for the cusp and with a second-order polynomial for the nightside (red curves), when considering energies and ratios in logarithmic scale. We can then correct the eVlasiator fluxes by multiplying them with the linear-scale equivalents of those analytical (polynomial) energy-dependent expressions, in the entire regions of interest (cusp and nightside oval). In other words, using the notations introduced in Section 2.3.2, we calculated the corrected eVlasiator differential number flux $\mathcal{F}_e^{\mathrm{VLAS,corrQ50}}(E)$ as:

$$\mathcal{F}_e^{\mathrm{VLAS,corrQ50}}(E) = \alpha_{\mathrm{day|night}}(E)\mathcal{F}_e^{\mathrm{VLAS}}(E) = 10^{\sum_i a_i (\log_{10} E)^i} \mathcal{F}_e^{\mathrm{VLAS}}(E), \tag{B1}$$

with $\mathcal{F}_e^{\mathrm{VLAS}}(E)$ the original eVlasiator differential number fluxes, $E$ the electron energy, and $a_i$ the coefficients of the fitted polynomials.

### B4 Adjustment based on the integrated energy fluxes

The median-based corrections presented above lead to an enhancement of the eVlasiator fluxes in a way which gives priority to increasing the energies needing it the most to better resemble observations. However, this increase is still insufficient to be representative of the energy input into the upper atmosphere associated with auroral electron precipitation as obtained in the DMSP

observations. Indeed, if we calculate the integrated energy fluxes along the satellite orbits for the corrected eVlasiator data, $\int \mathcal{F}_e^{\mathrm{VLAS}}(E)E\,\mathrm{d}E$, and compare them with the integrated energy fluxes measured by DMSP, $\int \mathcal{F}_e^{\mathrm{DMSP}}(E)E\,\mathrm{d}E$, we find that the former are still significantly lower than the latter. Taking the 90th percentile of the $\int \mathcal{F}_e^{\mathrm{VLAS}}(E)E\,\mathrm{d}E/\int \mathcal{F}_e^{\mathrm{DMSP}}(E)E\,\mathrm{d}E$ ratio along the DMSP orbits, we obtain values of 0.18 in the cusp and 0.24 in the nightside oval, which means that the corrected eVlasiator fluxes are still 4–5 times lower than in observations using this metric. Using the 90th percentile to compare

integrated energy fluxes was chosen to monitor for possible local overcorrection: we want to ensure that corrected eVlasiator fluxes are mostly on the order of or less than the DMSP fluxes, in terms of integrated energy flux. As we see here, performing the correction of the differential number fluxes based on median values yields significantly smaller integrated energy fluxes when compared to observations, meaning that the correction can be made stronger.

Therefore, we have adopted the following strategy to obtain corrected eVlasiator fluxes providing a good match with DM-
550 SP/SSJ measurements in terms of integrated energy flux: instead of taking median values of the DMSP/eVlasiator ratios to determine the analytical expression of the energy-dependent correction coefficients, we find the optimal percentiles of these ratios such that the integrated energy fluxes match as closely as possible between corrected eVlasiator fluxes and observations.

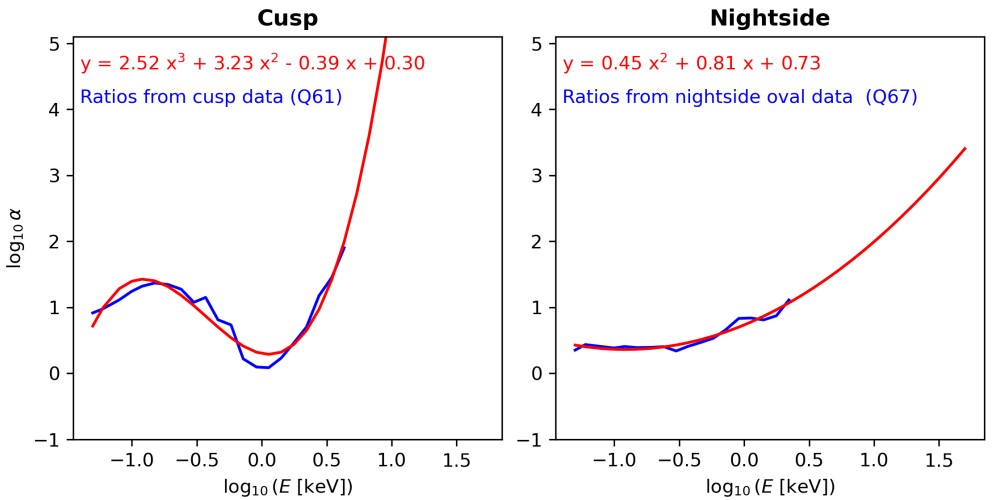

**Figure B5.** Same as Fig. B4 but using the 61st percentile of the DMSP/eVlasiator flux ratios for the cusp and the 67th percentile for the nightside fluxes.

We found that selecting the 61st (cusp) and 67th (nightside) percentiles of the DMSP/eVlasiator differential flux ratios along the orbits give the best results, with corrected eVlasiator precipitation being on par with DMSP/SSJ observations in
terms of integrated energy fluxes (90th percentile of the ratios of 0.99 and 1.03 for the cusp and nightside, respectively). The corresponding correction coefficients are given in Fig. B5.

Those coefficients were therefore retained for the eVlasiator flux correction, and produced the corrected fluxes shown in Fig. 3. Note that no extrapolation of the corrected fluxes at high energies (where the original eVlasiator fluxes are zero) is performed, meaning that the correction is only applied in the energy domain where the polynomial is fitted.

*Author contributions.* MB and MA developed the eVlasiator approach, with LK extending it to support 3D spatial meshes. MA ran the eVlasiator simulation presented in this work. MG developed the particle precipitation approach of Vlasiator, applied it to the eVlasiator data, conceptualised and implemented the ionospheric mapping and scaling approaches presented in the appendices and used in this study, and wrote the corresponding sections of the manuscript. MP is the Vlasiator PI, and she participated in conceptualisation of the study, and supervised the Vlasiator portion of the study. TH, MES, NK, and PTV participated in planning the WACCM-D simulations. TH carried out the WACCM-D simulations with support from MES. PTV produced the ionisation rates used in WACCM-D from the eVlasiator electron fluxes. TH analysed the WACCM-D data and led the writing of the paper, with all co-authors participating in discussions and providing feedback on the manuscript.

*Competing interests.* Minna Palmroth is a member of the editorial board of Annales Geophysicae.

*Acknowledgements.* We gratefully acknowledge the DMSP/SSJ community and data stewards and services for providing observations of precipitating particle fluxes. The work of MG is funded by the Research Council of Finland (grants 338629-AERGELC'H and 360433-ANAON). We acknowledge the Research Council of Finland grant 335554-ICT-SUNVAC, Finnish Centre of Excellence in Research of Sustainable Space (grant number 352846), and Flagship of Advanced Mathematics for Sensing Imaging and Modelling (grant number 359196).

Vlasiator development acknowledges European Research Council starting grant 200141-QuESpace and Consolidator grant 682068-PRESTISSIMO. The CSC–IT Center for Science in Finland and the PRACE Tier-0 supercomputer infrastructure in HLRS Stuttgart (grant nos. PRACE-2012061111 and PRACE-2014112573) are acknowledged as they made these results possible. The Vlasiator team wishes to thank the Finnish Grid and Cloud Infrastructure (FGCI) and specifically the University of Helsinki computing services for supporting this project with computational and data storage resources.

PTV, MG, and NK would like to thank the CHAMOS team for useful discussions (https://chamos.fmi.fi).

The scientific colour maps *batlow*, *imola*, and *lajolla* (Crameri, 2023) are used in this study to prevent visual distortion of the data and exclusion of readers with colour-vision deficiencies (Crameri et al., 2020).

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
