# Peer review of "Atmospheric odd nitrogen response to electron forcing from a 6D magnetospheric hybrid-kinetic simulation"

_Annales Geophysicae, 2024_

## Author Comment (AC1)

**Authors' response to referee comments on "Atmospheric odd nitrogen response to electron forcing from a 6D magnetospheric hybrid-kinetic simulation" by Häkkilä et al.**

Please find below our answers (in blue) to the comments (in black).

**Response to the comments of Referee #1**

The authors present an approach to use ionization rates derived from electron fluxes calculated by the eVlasiator magnetospheric model in the WACCM atmospheric model for to calculate the response of atmospheric nitrogen oxides and ozone to auroral electron forcing. Traditionally, ionization rates for such atmospheric modelling are derived in proxy-based parameterization so I think that this new approach would allow for a more physical input. As a disclaimer, I don't feel competent to judge the magnetospheric modeling part of the study, but feel more at home with the atmospheric modelling part. However, I see two major issues that I'd like the authors to consider before a publication can be considered regarding a) the motivation for this development, and b) the evaluation of the approach.

Response to general comments: We would like to thank the Reviewer for their expertise and the constructive comments. We appreciate the time devoted to the evaluation of our paper.

a) In the Introduction the authors state that "accurate modelling of the MLTI is required to [...] further our knowledge of the region" and that the complex dynamics lead to "great uncertainties in our understanding of the region". Furthermore it is stated that the lack of knowledge about "auroral electron forcing in climate models could [...] obstruct accurate evaluation of polar climate variability". While this may all be true, it is also very vague, and therefore it is difficult to judge whether the proposed new approach could help to generate new knowledge. What specifically is not known and why do the authors think that their approach can help to fill the knowledge gap? The abstract emphasizes the variability of the electron forcing on the hourly time scale. This point is not addressed in the manuscript anymore, but if it is the ability of eVlasiator to cover such time scales I think it would be necessary to argue why this is important for the stated purposes, e.g., the representation of polar climate variability. The WACCM results show that the eVlasiator electron fluxes lead to quite similar responses of atmospheric NOx and O3 as provided by the traditional Kp-based parameterization. What is the added value of the new approach? If a proper hypothesis about the benefits of the new approach is provided in the introduction, it can be tested and discussed in the manuscript. Instead, in the discussion, the authors put forward the "enhanced information on energy and spatial distribution" provided by the new approach. But if the expected benefits of this are not discussed it is difficult to judge whether the new approach is useful or not. The Discussion mentions that the approach would enable future "near-real-time predictions of the atmospheric response". Isn't this already possible with the existing parameterizations and the available Kp predictions?

Thank you for the feedback on the vague motivation of our paper. Our approach provides a unique methodology for the study of auroral electron precipitation. As pointed out by another Reviewer, there are existing electron forcing data that do cover the auroral electrons, but none specifically focused on the auroral part. We employ a novel method for the production of the precipitating auroral electron fluxes through the use of Vlasiator. Electron precipitation is known to be highly variable energetically, spatially, and temporally. While we do not currently cover the temporal variability of the auroral forcing, this is a first step towards the use of more accurate electron driving in atmospheric simulations. Incorporating several eVlasiator runs with this now-proven methodology could allow for time-variable driving analysis.
We would also like to point out that added value is provided through the agreement of the eVlasiator-derived and the parameterised auroral electron forcing. Since these data are somewhat in agreement, with similar scales and impacts, despite completely independent origins, this works to validate both approaches. This could also act as the basis for a more accurate parameterisation, e.g. using solar wind parameters to drive the ionisation rates in stead of (or in addition to) the $Kp$ index.

Since solar wind can be observed earlier than the geomagnetic activity used to calculate the $Kp$ index, this could lead to the improved, near-real-time predictions alluded to in the Discussion.

We will further address the vagueness of the motivation of our manuscript, both in terms of the Introduction and the Discussion, during the revision.

b) Somewhat related to point a), I'm missing an evaluation of the electron fluxes (or ionization rates) calculated by eVlasiator. This point is related to a) because if the goal is unclear, it is also unclear how to assess whether the presented new approach is useful to reach this goal. Regarding the atmospheric response, I think that the comparison of NOx responses produced by the new and an old approach, as provided in Section 3, is sufficient for the current study. However, the manuscript leaves me uncertain as to whether eVlasiator in its current form provides any useful information. Again, I'm happy to admit that this may be due to my limited knowledge of magnetospheric processes and modeling, but other potential readers from the atmospheric community may share this knowledge gap. In principle I don't see a problem in scaling the eVlasiator output with satellite observations. However, from Figures B1 to B3 in the Appendix, one might get the impression, that the eVlasiator output doesn't provide any similarity to the observations. Often, the observations and the model output differ by factors of several orders of magnitude. Is there any correlation between model and data? If I understand correctly, the magnetospheric simulation is run for fixed boundary conditions that represent a situation similar to that on the two days of the satellite overpasses. Why not check the quality of the scaling by comparing the scaled data with independent observations from a third day with similar conditions. Or if that is not possible, consider using only one of the days for scaling and check if the observations on the 2nd day are realistically reproduced. There may be many other ways to evaluate the approach, but without any evaluation I find it difficult to assess the usefulness of the approach.

Thank you for raising the point of the evaluation of the eVlasiator fluxes. In the revision, we will clarify that the goal is to demonstrate a new methodology to study the chemical response of the upper and middle atmosphere to particle forcing from space by introducing first-principle approaches to evaluate the precipitating electron fluxes used as an input for the atmospheric model (WACCM-D). The state-of-the-art is currently to use parametrised models, which may have limitations. There are two assets that eVlasiator brings to this aim. First, although it currently does not account for all the relevant processes leading to auroral electron precipitation, eVlasiator is a kinetic model of near-Earth space capable of producing a snapshot of electron parameters. Second, being a global model, eVlasiator provides such a snapshot in a global setup, meaning that the relevant parameters (the precipitating differential fluxes for our purposes here) are available as a 2D map above the polar regions.

The reason for scaling the eVlasiator fluxes with DMSP/SSJ observations is that, as mentioned above and in the manuscript, eVlasiator misses some of the important processes leading to electron pitch-angle scattering (e.g. chorus waves) and acceleration (e.g. field-aligned potential drops above the ionosphere). In our study, we therefore compare the eVlasiator fluxes with DMSP/SSJ fluxes during events for which the driving conditions (in terms of solar wind parameters) are similar to those used in our Vlasiator/eVlasiator runs. The comparison is carried out along the DMSP orbits, both on the dayside and on the nightside. Our aim is to make use of both the 2D extent of the eVlasiator fluxes and the "real-life" (though 1D) DMSP/SSJ

observations to produce a forcing dataset (electron fluxes) which can be deemed realistic given the driving conditions.

The reason why we have used the two events presented in the paper to determine the scaling coefficients is that real-life precipitation exhibits strong variability, up to orders of magnitude despite highly similar driving conditions. This is due, among other reasons, to the transient and explosive nature of substorms, which introduce both spatial and temporal variability in the precipitating particle fluxes. Therefore, it is expected that the eVlasiator fluxes scaled based on one DMSP overpass will still exhibit significant differences with the DMSP/SSJ fluxes measured during the second event. Since our objective is not to reproduce a single event but rather to evaluate the atmospheric response to a certain type of forcing conditions over long time scales (a year in our study), we prefer to derive the scaling coefficients for eVlasiator fluxes by combining the measurements from the two events for which we have DMSP data.

That being said – and this is something that is not discussed enough in the initial version of our manuscript – the scaling alone does not enable overcoming all the limitations of eVlasiator. Two aspects, in particular, remain: (i) the fact that the high-energy component of the precipitating spectrum (beyond the cutoff energy originating from the sparse description of phase-space density in eVlasiator) is still missing after the scaling. This is because the scaling is achieved by multiplying the original eVlasiator fluxes by the determined correction coefficient, and the resulting precipitating spectra are not extrapolated to higher energies. (ii) Currently, only one set of fluxes is available, as there is only one eVlasiator run, corresponding to a
single time step in the Vlasiator run. This limitation could be overcome by producing more eVlasiator runs, but the computational cost is very large (millions of CPUh). While introducing time variability in the forcing dataset would certainly be one of the next steps to improve the method, we prefer to keep things as simple as possible in the present study – which already introduces many elements of novelty.

Hence, in the revision, we will pay close attention to (i) clarify the objectives of our study, (ii) highlight the novel aspects
introduced by using eVlasiator fluxes as the forcing dataset, (iii) justify better the reasoning behind the choices made for their scaling with DMSP observations, and (iv) discuss more extensively and clearly the limitations of the method in its current version as well as future avenues for improvement.

In the following I will list further minor comments:

L26: "Hence, ... complex dynamics between the neutral atmosphere and the electromagnetic ionosphere". Maybe it is just the
wording, but what is meant, here? "Complex interactions? And why "hence"?

The purpose of this sentence was to act as a summary of the preceding introduction, reiterating that the polar MLTI is affected by both the atmospheric and the magnetospheric processes. We will be revising the Introduction, taking into account also the above general comment a), as well as feedback from the other Reviewers.

L39: "in long-term atmospheric and climate simulations although uncertainties are still present in the latter". Do you want to
say that only climate and not atmospheric simulations are uncertain?

Thank you for pointing out this inaccuracy. The uncertainty was in reference to there being more uncertainty in radiation belt electron data compared to solar energetic protons, though this is certainly not clear in the current text. The implication that there would be no uncertainty in solar protons data is also inaccurate. We will clarify and correct this in the revised version of the manuscript.

L41: "its impact on the stratospheric ozone balance is to a larger extent affected by polar atmosphere dynamics and is not fully understood". Almost nothing in our field of research is "fully understood". I think this is generally a poor motivation for a study. Please try to be specific on the knowledge gaps.

Thank you for pointing out the overly generic and vague nature of this statement. Specific deficiencies in our understanding of the role of the auroral forcing are already discussed on lines 54–58. We suggest to remove "and not fully understood" from
this sentence, and to slightly expand the discussion on lines 54–58.

Fig. 1 caption: "simulation of the 3D-3V magnetospheric simulation"?

Thank you for pointing this error out, the caption has been reworded in the revised manuscript.

Fig1: It's a beautiful figure, but it is hard to extract information from it. What is the color scale of panel a)? Also the grey grids are hard to identify.

We have added the colour scale (proton pressure) for Fig. 1a. The grey grids were chosen for the lack of better alternatives for showcasing the regions of high-resolution solution; Palmroth et al. (2023) includes more details on the underlying Vlasiator run.

Fig. 4: I'd prefer using the same size for all panels. With panel a) larger than the others I find it hard to identify differences, e.g., in "sharpness".

We will update Fig. 4 in the revised manuscript so that the panels will be the same size in order to facilitate more accurate comparisons.

Section 3.1.2, comparison of ionization rates from the parameterization and from VLAS: While I understand that there are differences I'd like to understand which of them are fundamental to the approaches. If, e.g., the missing "sharpness" of the oval is considered a weakness of the parameterization, is this fundamental or couldn't the parameterization be modified
accordingly?

Since the auroral electron fluxes from eVlasiator are currently based on a single time step, it is difficult to ascertain which features in the ionisation are characteristic of the physics-based simulation. Theoretically eVlasiator is able to produce the auroral electron fluxes (and thus ionisation rates) dependent on magnetospheric activity that the Kp-driven parametrisation cannot reproduce, since eVlasiator is physics-driven and more complex than the statistical model. This, however, requires
(e)Vlasiator runs under multiple different driving conditions. In our current paper we therefore limit the analysis to comparisons with the current parameterisation used in WACCM simulations.
Considering the computational requirements of running Vlasiator, we do not consider it unreasonable that the model might be used to adjust the existing parametrisation in the future. Vlasiator could be useful in creating parametrisations e.g. driven by solar wind parameters, but in order to facilitate such developments, the necessary connections must first be understood.

L294: "realistic auroral electron precipitation fluxes from eVlasiator". This relates to my main point b). "Realistic" by which metric?

The ionisation rates are actually derived from precipitating electron fluxes which are driven by true *first-principles* physical simulations of magnetotail reconnection. We appreciate the need to justify the assertion that the fluxes provided by eVlasiator are "realistic" and we will reevaluate the wording of this statement in the revision.

L303 "Our results also indicate that the current parameterization of aurora may be overestimating the auroral forcing". Yes, may be, but later it is stated that "eVlasiator [...] underestimated the total precipitating fluxes. Is there a reason to believe that the first effect dominates?

These two statements refer to different aspects. The first one is a conclusion of our study, based on the utilisation of eVlasiator fluxes *scaled with DMSP observations* as inputs for electron forcing for WACCM-D. The second one is a discussion about
why scaling eVlasiator was necessary, since the model does not include all relevant processes leading to auroral electron precipitation. We will make this difference appear more clearly in the revised manuscript.

L337: "for the detailed study of solar wind – atmosphere interaction". Besides the main issue a) which applies also here, because I find this very vague, I'd also suggest to reconsider the use of the word "interaction", which I usually understand to work in two ways. However the coupling of the magnetospheric and atmospheric models suggested here is clearly one-way,
so it would only allow to study effects of auroral electron forcing on the atmosphere. I know that publications in the larger field often use the term "solar-terrestrial interactions" or similar, but I'd prefer an unambiguous use of the language.

Thank you for pointing out that while "interaction" may be a commonly used term, it is indeed ambiguous. It is true that the interaction presented in our study is strictly one-way, and as such we suggest to replace the sentence as follows: "For the future, this work paves the way for a more complete description of auroral electron forcing in atmospheric simulations. Eventually this would enable the detailed study of how the solar wind affects the atmosphere through particle precipitation." We will further address the vagueness of the Conclusions during the revision.

---

## Author Comment (AC2)

**Authors' response to referee comments on "Atmospheric odd nitrogen response to electron forcing from a 6D magnetospheric hybrid-kinetic simulation" by Häkkilä et al.**

Please find below our answers (in blue) to the comments (in black).

**Response to the comments of Referee #2**

Review of "Atmospheric odd nitrogen response to electron forcing from a 6D magnetospheric hybrid-kinetic simulation" by Hakkila et al.
This study applies the WACCM model to simulate the effects of energetic electron precipitation in the Earth's thermosphere, mesosphere, and upper stratosphere, focusing on the impact on NOx and O3 levels. The study analyzes model runs with electron precipitation derived from magnetospheric hybrid-kinetic simulations using the combined Vlasiator and eVlasiator
model framework. Additionally, comparisons are made with WACCM results incorporating simplified, nominal auroral precipitation maps parameterized by the geomagnetic Kp index (ranging from 0 to 5, with 0 indicating no auroral inputs). The findings indicate that auroral electron precipitation significantly enhances NOx concentrations during the polar winter, while its impact on upper stratospheric O3 is negligible. This study is very interesting, however, the following concerns need to be addressed.

Response to general comments: We thank the Reviewer for the insightful comments and questions regarding our manuscript. We appreciate the time taken to review our paper. We would like to point out a possible misunderstanding concerning the Kp parameterization. Setting the $Kp$ index to 0 does not result in no auroral forcing in the parameterization, as can be seen in Fig. 4b. For this reason a separate reference run (REF) was necessary for comparisons in our study.

My biggest concern is that the electron precipitation estimated using Vlasiator-eVlasiator to drive the WACCM model seems
to be fixed, not only spatially (precisely, in the geomagnetic coordinate system) but also temporally. This simplification significantly diminishes the potential benefits of using the sophisticated hybrid-kinetic simulation of particle precipitation, as compared to the simpler parameterization using the Kp index. Apart from a brief comparison of ion production rates in Figure 5, the study lacks detailed discussions that would adequately justify the preference for hybrid-kinetic simulations. To convincingly demonstrate the necessity and value of these complex and computationally intensive simulations, more
comparisons and in-depth analyses should be conducted.

Electron precipitation is known to be highly variable energetically, spatially, and temporally. Our approach provides a unique methodology for the study of this auroral electron precipitation, as we employ a novel method for the production of the precipitating auroral electron fluxes through the use of Vlasiator. The Reviewer is correct in that this does not currently cover the temporal variability of the auroral forcing, but it does provide more information on the spatial and energetic distribution of
the auroral electrons. This is a first step towards the use of more accurate electron driving in atmospheric simulations. Incorporating several eVlasiator runs with this now-proven methodology could allow for time-variable driving analysis, and in the future this could be used to produce more accurate parametric forcing of the polar MLTI.
We will add discussion on the ultimate usefulness of the the use eVlasiator in the revision. We will also expand the analysis of the ionisation rates in Section 3.1.2, and include analysis of momentary ionisation rates in addition to the daily averages
currently shown in Figs. 4 and 5.

Clarifications regarding the Vlasiator-eVlasiator simulation are needed.

The proton-electron mass ratio is inconsistent between line 97 and line 115.

The mass ratio used in the 2D simulation in Alho et al. (2022) was larger than in the 3D simulation presented in this manuscript. The change was made to reduce computational costs. Differences between runs with different mass ratios were observed to be small, consisting of minor shifts in spatial structures.
We will clarify the manuscript to remove this confusion.

Lines 110-111: The upstream solar wind speed and electron temperature are significantly higher than typical values, while the solar wind density is considerably lower. Note that the obtained simulation results are used to drive the WACCM model over an extended period to represent typical precipitation conditions near Kp=1-2. Therefore, these settings are highly inappropriate.

We acknowledge that the driving conditions for the Vlasiator simulation do not match typical driving conditions. They represent conditions which are found during fast solar wind conditions, with a reasonable fast solar wind speed ensuring expedient initial condition formation, southward IMF to trigger efficient magnetopause reconnection driving global dynamics, and a low solar wind density in order to ensure the ion skin depth is similar to comparison 5D Vlasiator runs. A global 6D Vlasiator simulation requires significant computational resources to complete, and thus, it was not feasible to perform a new simulation (beyond the also-expensive eVlasiator run) for this purpose. As mentioned in Sect. 2.4.2, the driving conditions during the two DMSP overpasses used for scaling eVlasiator fluxes were associated with $Kp$ values of 2 and 3.

Does eqn (1) specify the precipitating flux along the local magnetic field direction? Are magnetic mirroring effects considered?

The precipitating flux along the local magnetic field direction is specified by eqn. (1), and magnetic mirroring effects are considered. The bounce loss cone referenced on line 104 is the loss cone resulting from magnetic mirroring.

In Vlasiator, are electrons treated as a massless fluid?

Yes, in Vlasiator electrons are a massless charge-neutralizing fluid. Though considered massless and ensuring charge-neutrality, in MHD Ohm's law, the electron pressure gradient term is also included, assuming that inflow boundary electrons have equal temperature and density to solar wind ions, and that they evolve in an adiabatic fashion in the magnetospheric domain. This is a simplification, and the electron pressure term has only a minor contribution to global dynamics.

What are the boundary conditions near 4.8 RE? Do the models account for varying ionospheric conductances?

In the presented Vlasiator run, the inner boundary is a near-perfect conducting sphere with copy conditions for ion VDFs. A new ionospheric inner boundary, considering ionospheric conductances and field-aligned currents, is introduced in Ganse et al. (2024, https://doi.org/10.5194/gmd-2024-101). Using this model as a basis for eVlasiator runs in the future would be an interesting avenue of investigation.

The dates used for eVlasiator-DMSP calibration in line 135 are not consistent with the date used for magnetic field line mapping in line 356.

As explained in Appendix A, the date used for magnetic line mapping with the Tsyganenko 2001 model was selected to match equinox conditions, during which the geomagnetic dipole was almost perpendicular to the ecliptic plane. The reason behind this choice is that the used run of Vlasiator (and hence eVlasiator) for this study uses an untilted geomagnetic dipole. The geomagnetic activity level used as an input for T01 was chosen to be represented by a Dst value of –30 nT, consistent with mildly enhanced geomagnetic conditions as seen in the two DMSP overpasses. Note that our goal here is not to reproduce a real event, but rather to demonstrate a methodology, which is why we selected a parametrisation of the field line mapping as close as possible to the hypotheses (dipole alignment, driving conditions) used for the Vlasiator and eVlasiator runs.

Under the assumption of the non-tilted dipole magnetic field, the north-south asymmetry (in MLT-MLAT) is generally neglected, as seen in Figure 3. Explicitly clarify this in the text.

We will state explicitly in the text that lack of a dipole tilt will drastically reduce north-south asymmetry.

I don't understand how the calibration after the DMSP data along specific orbits can be useful, in physics. In particular, note that the calibration is temporally independent, while the model-data disagreement varies with time. Therefore, I suspect that even after applying the proposed calibration as in Figure B5, the ratio plots (similar to panels e-f in Figure B1 and Figure B2) will still show discrepancies of orders of magnitude.

It is inevitable that significant discrepancies still exist between the forcing data set (scaled eVlasiator precipitating electron
fluxes) and the real measurements by DMSP. Indeed, as can be seen by comparing the DMSP observations shown in Figs. B1 and B2, the two events are characterised by order-of-magnitude differences, underlying how variable auroral precipitation can be despite similar driving conditions.
That being said, our aim in this study is not to obtain estimates of the atmospheric chemistry response to a given event, but rather to assess whether the long-term (over one year) forcing of the atmosphere by auroral electron precipitation using the
Kp-based parametrisation of the fluxes can be improved by employing first-principle-based electron flux inputs. In that sense, scaling the eVlasiator fluxes based on two DMSP overpasses rather than only one helps in taking into account the high variability of electron precipitation. We do not expect to obtain a perfect match between the scaled eVlasiator fluxes and a given DMSP event, but we nonetheless try to reduce the discrepancy existing without the scaling (due to not all the processes leading to electron precipitation being present in eVlasiator).
While the results obtained in our study still have lots of room for improvement and have many limitations, we still want to emphasise that they are well beyond the current state-of-the-art, in the sense that eVlasiator is the first global model of near-Earth space based on solving the Vlasov equation for electrons and providing differential number fluxes of auroral electron precipitation. Further developments and improvements are naturally needed, but this study provides an initial evaluation of the benefits and shortcomings of our method to evaluate the atmosphere's chemical response to particle forcing
associated with geomagnetic activity.

There are a few issues in atmospheric ionization calculations.

Line 176, the parameterization method of electron impact ionization is not applicable to incident energies below 100 eV. Nevertheless, it is adequate to consider only the incident energy range of 100 eV to 50 keV in this study, as <100 eV electrons do not penetrate below 140 km. See Fig 6.10 in Fang [2022], "Chapter 6.2—Fast calculation of particle impact ionization
from precipitating energetic electrons and protons in the earth's Atmosphere"
(https://doi.org/10.1016/B978-0-12-821366-7.00005-6)

Thank you for pointing this out. Indeed, spectral energy range of 50 eV to 50 keV is used in the ionisation rate calculation. The energy grid has 32 log-spaced points most of which can be seen in Fig. 2. Three of these energy grid points are below 100 eV. We have now removed those three points and calculated the atmospheric ionisation rates again. At the model altitude
range considered here, i.e. below 140 km, the removal of <100 eV electrons from the spectrum does not change ionisation. As pointed out by the Reviewer, electrons with such low energy cannot penetrate below the model's upper altitude limit. We have also re-examined the ionisation rates at the higher end of the auroral electron spectrum. Since the medium-energy electron (MEE) forcing in CMIP6, which was used in our WACCM simulations, already accounts for electron forcing at energies greater than 30 keV, we have also removed the eVlasiator derived auroral electron fluxes at energies 30–50 keV. This avoids the possibility that the eVlasiator auroral forcing and the MEE forcing might partially overlap. The eVlasiator derived electron fluxes at these energies were so small, that this removal also had no significant impact on the results of our study. The final energy range of the eVlasiator-derived auroral electron input used in our WACCM-D simulation is 100 eV – 30 keV. As explained above, the adjustment of the auroral electron precipitation spectral energy range has negligible impact on the calculated ionisation rates and corresponding atmospheric response. We will add a brief discussion about this issue to

Section 2.4.1 of the manuscript.

Line 177, the use of the NRLMSIS model for the ionization calculation introduces inconsistency with the WACCM-specified neutral profiles, especially for energetic electrons penetrating below 140 km. Are the resulting ionization rates above 140 km, as shown in Figure 2b, disregarded in the WACCM runs? If so, why not use the WACCM atmosphere directly to calculate the ionization?

The ionisation rate calculation requires an atmosphere which for the calculation was taken from the NRLMSISE-00 model. However, according to the CMIP6 procedure (Matthes et al., 2017), the ionization rates are then divided by the MSIS mass density which effectively removes the atmospheric "signature". When the rates are used in WACCM, they are multiplied by the WACCM mass density profiles making the forcing consistent with the WACCM atmosphere.
We will note this in Section 2.4.1 of the manuscript.

Other minor comments:

Line 13, and throughout the paper, delete the space between numerical values and the percentage symbol (%).

We have used a space preceding the percentage symbol (%), as instructed in the International System of Units (9th edition, Section 5.4.7, https://www.bipm.org/en/publications/si-brochure).

Line 19, delete "unique", or change to "particular"

We have changed this in the revised manuscript.

Line 30, change "solar energetic radiation" to "solar radiation"

We have corrected this in the revised manuscript.

Line 36, "There are three primary EPP sources of NOx"

We have changed this in the revised manuscript.

Lines 36-37, change "solar protons" to "solar energetic protons"

We have changed this to "solar proton events" in the revised manuscript.

In addition, what about "solar energetic electrons"?

Solar wind electrons do not have enough rigidity (momentum per charge) to penetrate the Earth's magnetosphere and directly enter the atmosphere, unlike solar wind energetic protons which can enter the atmosphere at polar latitudes during big solar
storms (referred to as solar proton events). Solar wind electrons are captured by the magnetic field of the Earth and stored e.g.

in the Van Allen radiation belts. The auroral electrons and the radiation belt electrons (or medium-energy electrons) therefore account for the precipitation of these electrons into the polar atmosphere. Though solar in origin, they are typically not referred to as solar energetic electrons, since they do not precipitate directly into the atmosphere but only after being energised by magnetospheric processes during magnetic storms.

Lines 42-43, rephrase the sentence. Auroral precipitation is a continuous phenomenon that occurs not only during substorms and has sources beyond just the magnetotail.

Thank you for pointing this out; we propose to rephrase this statement as follows in the revised manuscript: "Auroral electrons, with typical energies on the order of a few kiloelectronvolts, precipitate from the magnetosphere into the upper atmosphere in the polar regions, particularly along the auroral oval located most of the time between 60 and 75° geomagnetic latitude. While auroral precipitation occurs on a continuous basis, its flux is significantly enhanced during magnetospheric substorms, when the magnetotail is suddenly disrupted and launches a large number of electrons (and protons) of variable energies towards the ionosphere (Palmroth et al., 2017; Palmroth et al., 2023)."

Line 70, briefly explain "6D" (space and velocity space)

We have corrected this in the revised manuscript.

Line 74, what does "Cartesian 2D" refer to?

The mesh is discretised in Cartesian coordinates, and the spatial grid is 2-dimensional, the out-of-plane dimension having a thickness of a single cell only.

Line 81, what does "fields" refer to? It can easily be confused with "magnetic field" and "electric field".

This indeed refers to both electric and magnetic fields. This has been clarified in the manuscript.

Line 86, the use of "at full strength" is not appropriate.

We have changed this to refer to actual strength. The reason this statement is included is to clarify that a reduced dipole strength, sometimes implemented in other global magnetospheric models, has not been used here.

Figure 1a, what does the color represent?

The colour represents proton pressure. We have updated Fig. 1a to include a colour scale and revised the caption.

Figures 1c-1e, what are the relative locations of the three points? Are they along one specific open magnetic field line? Why are the velocity space plots not organized in terms of parallel and perpendicular velocities? What findings are drawn from the comparison among these three plots?

The selected panels showcase the diversity of electron distribution functions found along those field lines which facilitate precipitation. They are provided for instructive purposes only. Maintaining the VDFs in Cartesian coordinates and indicating the magnetic field direction with an arrow showcases that they are from significantly different positions along the curved magnetic field line.

Line 128, swap the order of the two processes to align with the two items listed earlier in line 127.

Thank you for this suggestion; we will make the change in the revision.

Line 167, the sentence is confusing. What is the relationship between the NOEM-specified NO and the precipitating electron induced NOx at 140 km altitude? Is there any inconsistency here?

NOEM is used to set the upper boundary condition of NO concentration in WACCM simulations. This accounts for the production of NO at altitudes above WACCM's altitude range, and its exclusion would lead to a lack of NO at high altitudes. The use of NOEM in WACCM makes it necessary to set the $Kp$ value for the VLAS simulation, as explained in Section 2.4.2 of the manuscript. We will try to clarify this further in the revised manuscript.

Lines 171-172, what is the relationship between "particle impact ionization" and "dissociative ionization"? What is "secondary electron dissociation"?

Particle impact ionization creates an electron-ion molecule pair (e.g. e and $N_2^+$), dissociative ionization also separates the ion molecule into an atomic ion and and a neutral atom (e.g. e, $N^+$, N). Further, if the created electron has enough energy, it can dissociate molecules itself. A review of these is given by Sinnhuber et al. (2012, https://doi.org/10.1007/s10712-012-9201-3).

Line 175, delete "temporal"

We have corrected this in the revised manuscript.

Line 185, briefly specify the recommended forcing conditions so that readers do not need to refer to the reference to understand the driving conditions.

The recommended CMIP6 solar and geomagnetic forcing, as used in this study, includes total and spectral irradiance as well as atmospheric ionisation rates resulting from solar protons events, medium-energy electrons, and galactic cosmic rays. This brief introduction is given in Section 2.4 (lines 162–166) of the manuscript. To avoid possible confusion and unnecessary repetition, we suggest restructuring Section 2.4, including its subsections, in the revised manuscript such that: (i) All descriptions of the WACCM simulations are combined into a single subsection 2.4.1, and (i) the production of the ionisation rates into subsection 2.4.2.

Line 195, I don't understand this sentence.

We suggest to reformulate this sentence in the revised manuscript as follows: "The eVlasiator-derived auroral electron forcing covers the full 24 hours of MLTs, and this forcing is repeated every day of the WACCM-D simulation."

Line 197, delete "and no Kp driven parameterized aurora", which is redundant

We have corrected this in the revised manuscript.

Line 197, it is my understanding that the REF run excludes auroral precipitation, but still includes SEP impact, according to line 263?

This is correct. All the WACCM-D simulations, including REF, include the recommended CMIP6 solar proton and medium-energy electron forcing. We have clarified this in the revised manuscript.

Line 206, how is the Kp index used to drive the VLAS run?

As stated in Section 2.4 (lines 166–168), NOEM uses the $Kp$ index to determine NO at top of WACCM's altitude range. More precisely, WACCM makes use of NOEM to set the upper boundary condition of NO concentration. It is therefore necessary to set the $Kp$ index for the VLAS simulation as well.

Line 220, I understand that these ionization rates are used to drive the WACCM model, not "from" the model.

Thank you for pointing this out. While the Kp-driven auroral electron forcing is calculated within WACCM based on the input Kp index, this is ultimately true. We have removed "from WACCM-D" from the title of Section 3.1.2 of the revised manuscript.

Figure 4, I don't see how altitude-integrated ionization rates can be useful. As dissociative recombination rates are altitude dependent, the efficiency of ionization in converting into ion/electron density increase also varies with altitude. This makes the integration of the ionization rate over altitude not meaningful.

The altitude-integrated ionisation rates are a measure of the total NOx production. NO production has been shown to approximately correspond to integrated ionisation rates with a factor of around 1.25, see, e.g., Nieder et al. (2015, https://doi.org/10.1002/2013JA019044) and Kirkwood et al. (2015, https://doi.org/10.5194/angeo-33-561-2015). It is true that the electron density resulting from the auroral ionisation is affected by the dissociative recombination. The NO density, however, is also affected by the recombination due to the balance of chemical production and loss, and transport. Nevertheless, we will update Fig. 4 to include the ionisation rates at select altitudes to better illustrate the vertical variability.

Line 233, what does it mean by "the lower boundary of the parameterization"?

This refers to the fact that WACCM's Kp parameterization of auroral electron forcing does not extend below about 95 km in altitude. We will clarify this in the revised manuscript.

Line 237, the polar cap excludes the auroral oval. You may want to change "polar cap averaged" to "polar averaged"

We will change this during the revision.

Line 269, where are the "troughs"? I cannot find them in Fig 7d.

The troughs are in reference to the upper stratosphere, depicted in Fig. 7f for the NH. Admittedly there is only one trough, not multiple. We will rephrase this to be more accurate in the revised manuscript.

Line 283, change "showing" to "due to"

We have changed this in the revised manuscript.

Line 293, change to "our results demonstrate the coupling between the magnetosphere and the atmosphere through electron precipitation", or something similar.

We will change this in the revised manuscript.

Line 303, change "at least" to "likely"

We believe that this would alter the meaning of the sentence, and cannot oblige in this instance.

Line 307, insert "be" prior to "considered"

We have corrected this in the revised manuscript.

Line 351, change "seed points" to "start points"? The word "seed" implies sources.

We will make this change in the revised manuscript.

Figure A1 caption, what are the thoughts behind the use of "7.5 RE"? In addition, change "Cartesian in MLAT-MLT" to "regularly spaced in MLAT-MLT".

The value of 7.5 $R_{\mathrm{E}}$ used for field line tracing was empirically determined. Essentially, it ensures that this distance is sufficient to reach the transition region for all the closed field lines on the nightside in this run without extending unnecessarily far down the magnetotail or in the cusp for the open field lines. We will add this justification in the revised manuscript.

Figure B1 caption, "DMSP/SSJ (contour lines)"

We will make this addition in the revised manuscript.

Line 379, briefly explain why the high-energy component is missing

This is due to the sparsity threshold used in eVlasiator simulations, which discards velocity cells within which the phase-space density is below the threshold to keep the simulation computationally feasible. Since the phase-space density
decreases near the edges of the velocity distribution, applying the sparsity threshold creates a sharp drop at those edges, which translates into a cutoff at high energies in the precipitating flux.
We will add this precision in the revised manuscript.

Line 399, delete "el cm^-2 s-1 sr^-1 eV^-1"? I think this is the ratio, not flux, that you are talking about.

Thank you for noticing this mistake; we will correct it in the revised manuscript.

Line 404 and throughout the paper, change "quantile" to "percentile"

Thank you for pointing out this language inaccuracy; we will replace "quantile" with "percentile" throughout the revised manuscript.

---

## Author Comment (AC3)

**Authors' response to referee comments on "Atmospheric odd nitrogen response to electron forcing from a 6D magnetospheric hybrid-kinetic simulation" by Häkkilä et al.**

Please find below our answers (in blue) to the comments (in black).

**Response to the comments of Referee #3**

5    Review of "Atmospheric odd nitrogen response to electron forcing from a 6D magnetospheric hybrid-kinetic simulation" by Häkkilä, T. et al.
The manuscript presents a novel approach to combine magnetospheric modelling with whole-atmosphere modelling to investigate the auroral production of odd nitrogen. The magnetospheric model drives the precipitating electrons, and the whole-atmosphere model WACCM is used for the production of the odd nitrogen species, NO in this case.

10   The study is well thought out and an interesting approach that could shed more light on current discrepancies between measurements and current climate model simulations of the lower thermosphere. Thus it would be a worthy contribution for Ann. Geophys. However, I could recommend publication after minor revision of the manuscript. I feel some of the arguments need to be improved and made more clear, some of the issues might be caused by the language used and might be resolved with a slight rewrite or rephrasing.

15   Response to general feedback: We would like to thank the Reviewer for their positive comments and the incredibly detailed feedback. We appreciate the time devoted to the evaluation of our paper.

General comments
1. I understand that for numerical reasons the electron mass is different from the real value. However, this changes also the (kinetic) energy for the same speed, or the speed for the same kinetic energy. The energy and energy flux affect the
20   precipitation characteristics, and the velocity is part of the Vlasov equation. How are these corrected back to realistic values corresponding to real electrons? Apparently the mass used is roughly 46 times the real electron mass, which would results in almost a factor of 7 for the velocities. Although [1] briefly discusses the impact of the different mass, the main arguments and justification could be stated here to justify the choice.
In addition, the changed electron mass is given relative to the proton mass, I suggest the authors also state it relative to the real
25   electron mass, maybe denoted by me0 or similar, like it is done in [1]. I think it would make it clearer that it is the electron mass that is scaled and not the proton mass.
Also, currently there are two values listed, 5.11 MeV, citing [1], and what amounts to roughly 23.46 MeV a bit further down in the text. Which one was used? I suggest to remove the other to avoid confusion.

We have clarified the discussions on the electron mass in the manuscript. In response to the specific points, the choice of the
30   mass ratio does not affect the evaluated differential number fluxes (Eq. (1)). The mass ratio in the 2D run in [1] was indeed larger than in the 3D eVlasiator run shown here. The change for the 3D simulation was made to reduce computational costs. Differences between runs with different mass ratios were observed to be small, consisting of minor shifts in spatial structures. The change in the mass ratio is now explained explicitly in the manuscript.

2. I see that the authors use classical quantities for the kinetic energy (after Eq. (1)). [1] lists the range of velocities up to
35   42000 km/s per dimension, which leads to a maximum speed of about 73000 km/s, or roughly 25% light speed. How large is the error introduced here by using the classical approximation instead of the relativistic description? Why not use relativistic quantities throughout? At least some of the impacts and reasons should be discussed briefly (best with a reference).
A related question is why Vlasiator uses velocity and not the momentum to model the phase space more directly? One could then use for example the Liouville theorem to constrain the solutions further. I am not suggesting to rewrite Vlasiator for the
40   current study, but the authors and model developers might want to think about that for future developments.

We would like to emphasise that the maximum velocity space extents are not actually modelled (due to our sparsity approach), and no particle velocities actually modelled will ever even reach the single-dimension maximum value. In [1] some particles do travel at over 10,000 km/s which indeed goes into the relativistic regime, but in the current study, due to more massive electrons being used, velocities remain lower (as shown in panels c,d,e of Fig. 1), and a non-relativistic solution is considered sufficient.

We thank the Referee for an interesting point of discussion. Using momentum directly in Vlasiator is an interesting suggestion and has been investigated in the past. It would not be completely impossible, but would require a re-write of the whole simulation model, as the SLICE-3D decomposition of acceleration into three shear motions would need to be also made relativistic, and this is not straightforward in the least. As regular Vlasiator simulations are constrained to the non-relativistic regime, this has thus not been a worthwhile avenue of code development.

3. Please list the absolute changes in addition to the relative ones, e.g. as 200% (from x to y). That would make it easier to judge if the change is from 1 to 2 (or 3?) molecules, or from $10^8$ to $3 \times 10^8$. If not in the abstract, then at least in the results and conclusions sections.

We will add the changes in concentrations in absolute values to the Results and Conclusions wherever appropriate. For example, for the Abstract, the approximate NOx changes in the SH are 200 % (from $1.6 \times 10^{14}$ to $5.1 \times 10^{14}$ molecules per cm$^2$), 50 % (from $3.1 \times 10^{14}$ to $4.9 \times 10^{14}$ molecules per cm$^2$), and 7 % (from $1.6 \times 10^{15}$ to $1.7 \times 10^{15}$ molecules per cm$^2$) in the lower thermosphere, mesosphere, and upper stratosphere, respectively.

4. I understand that maybe due to limited resources, only single WACCM runs have been performed and not ensemble runs. However, I did not find that mentioned anywhere in the manuscript. Please add some words of discussion, and maybe add it to the "next steps", as only this would allow to assess the quality of the resulting NOx values.

Since we use specified dynamics in our simulations, below about 60 km an ensemble run would show no differences between the members. At altitudes above, based on our previous work, we found that depending on the season of the year NO variability between ensemble members can be up to 5 % near 80 km (NOx minimum) and smaller at other altitudes. Figure S1 shows the variability of a 10-member WACCM ensemble using specified dynamics. The data in Fig. S1 is from the simulations used in Verronen et al. (2020, https://doi.org/10.5194/angeo-38-833-2020), though this figure does not appear in the paper.

Since our WACCM simulations show more deviation in NOx from the reference than the previously determined 5 %, especially at thermospheric and mesospheric altitudes, we believe the results to be valid without the need for ensemble runs. We will add a brief discussion on this to Section 3.2 of the revised manuscript.

Specific comments

L. 13: See general comment #3, absolute changes would help.

We will add the absolute changes to the Results and Conclusions in the revised manuscript.

L. 22–23: "The polar MLTI depends on solar radiation" is a bit vague. Please revise to make it clearer to what aspect of the MLTI the authors refer to.

We will clarify the Introduction to address the vagueness during the revision, also taking into account the feedback from the other Reviewers.

[Figure]

**Figure S1.** Standard deviation (SD) of the Southern Hemisphere (SH) polar-cap monthly O3, NOx, HOx, T, Ox, and HNO3 anomalies from a 10-member ensemble of SD-WACCM-D APEEP REF simulations, relative to the ensemble mean. Plus signs (+) indicate SDs for the individual months of January, February, March, October, November, and December (summer); circles are for individual months from April to September (winter); black lines are the minimum and maximum SD at each pressure level; and the red thick line is the median of all monthly SDs. See Verronen et al. (2020) for ensemble description.

L. 48: "... simplistic realization ..." only applies to the CMIP6 setup. There are other parametrizations available (e.g. [2]), which have been compared to each other and to measurements (e.g. [3]; [4]). However, so far not explicitly focussing on the auroral part.

80   Thank you for pointing this out; we will add discussion to the Introduction on existing approaches to auroral electron forcing in atmospheric simulations. As pointed out by the Reviewer, most of these methods do not specifically focus on the auroral part of the electron precipitation. This is the novelty of our study: we present a new methodology for the production of the precipitating electrons at auroral energies, and explicitly evaluate the atmospheric impact of the auroral forcing.

L. 75–76: In other words, the "6D" space is what is also known as the "velocity phase space".

85   There are many different ways of phrasing this, but yes, we model six-dimensional phase-space density in three spatial and three velocity dimensions. The manuscript has been amended on line 70 to reflect this.

L. 81: What are the "fields" here? Is this solution obtained iteratively? Please be more specific.

The manuscript has been amended to describe how we model both electric and magnetic fields. Our field solver approach is based on a finite volume approach described in Palmroth et al. (2018), and a reference was added here.

90    L. 84: What is "the ion inertial length", can the authors give specific numbers here?

The ion inertial length $d_i = c/\omega_{pi}$ where $c$ is the speed of light and $\omega_{pi}$ is the ion plasma frequency. This varies as a function of ion density, so a single value for the whole magnetospheric domain cannot be provided, but values can be of the order of, e.g., a hundred km.

L. 86: Which magnetic field model was used?

95    As the Vlasiator domain inner boundary is at nearly 5 RE, a simple dipole moment is sufficient. This has been clarified in the manuscript.

L. 95: What are "connected field lines"? Aren't field lines "connected" by definition? Maybe just "along field lines" is enough.

Field lines which connect back to the ionosphere without reaching the Vlasiator computational domain (in equatorial regions) are not connected to the Vlasiator domain. However, we agree that this distinction is not necessary, and have removed that
100   word.

L. 96: See general comment #1, please state the actual value used in the study and explain why it was chosen.

We have updated the manuscript to clarify this.

L. 100: Please add $\mathcal{F}_e$ to the text: "... number flux $\mathcal{F}_e$ at position $r$ and energy E is given by ..."

We have made this correction.

105   L. 104: Which quantities are averaged? From reading Eq. (1), this would be $\theta$ and $\phi$. If this is correct, please add it to the description.

We have updated the manuscript to clarify this.

L. 110: How typical are these conditions? Was the driving force applied constantly over the whole simulation period, or intermittently to simulate reality? Please explain the rationale for choosing those numbers.

110   These driving conditions are constant throughout the Vlasiator simulation, which has been added to the manuscript. These conditions depict fast solar wind stream conditions, ensuring fast initialisation and more simulation time for actual global dynamics modelling. Dynamic driving conditions are supported by Vlasiator, but were not included in this run, as each new driving condition takes time to propagate throughout the magnetospheric domain, requiring significant computational resources to model.

115   L. 115: Here, suddenly $m_p/m_e$ is different from what was described before, please make these numbers consistent.

The manuscript has been amended to clarify this.

L. 129: "... fluxes ... are lower than ..." is difficult to assess without knowing the true flux. I suggest to use a more careful wording: "... fluxes ... might differ from reality."

*Thank you for this suggestion; we will make the change in the revised paper.*

120 L. 131–132: In my opinion, the authors are stretching the notion of "calibration" here. Typically instruments are calibrated by comparing to a measurement standard to adjust the scale over a wide range of values. Here, only two days with the same conditions are compared. A few questions arise, first, are these typical conditions, in the sense that they occur very often? Second, how would this comparison hold up against a third day with similar conditions, and third, how would it hold up in a comparison with different driving conditions? Ideally, a "calibration" would have to be carried out based on the average fluxes

125 (from SSJ or any other source) during different driving conditions. Unless this is fixed, I suggest to use a better terminology, maybe "adjustment" instead of "calibration". At least another comparison on a different day should be checked if the author's adjustment holds up.

*Thank you for pointing this out. We agree that the term of "calibration" was poorly chosen. We will replace it with "scaling" in the revised manuscript.*

130 *The driving conditions corresponding to the Vlasiator (and eVlasiator) runs are not very typical, and correspond to a solar wind high-speed stream with a particularly large solar wind speed and low density. These driving parameters are chosen as a result of the computational constraints associated with running Vlasiator.*
*Our goal here is primarily to present a new methodology to evaluate the effects of auroral electron precipitation on the upper-atmospheric chemistry. We want to investigate to what extent using inputs for WACCM-D (i.e. precipitating electron*

135 *fluxes) coming from first-principle simulations of the near-Earth plasma environment differs from using a Kp-driven parametrisation. We are aware that our method suffers from multiple limitations, such as the fact that, at the moment, we can only use one set of electron fluxes from a single time step simulated with eVlasiator. However, the Kp-based parametrisation also consists of a single set of forcing fluxes as a function of geomagnetic latitude, magnetic local time, and electron energy. In that sense, comparing the chemical response of the upper atmosphere to the forcing from eVlasiator to that from various*

140 *Kp values in the parametric approach already provides valuable insight, although the eVlasiator fluxes (after scaling with DMSP observations during similar solar wind driving conditions) cannot realistically aim at fully reproducing real events. Therefore, while there are still many further developments needed to reach a point where eVlasiator fluxes (scaled with observations or not) can be readily used to study atmospheric chemistry under auroral electron forcing, this paper presents the first step towards this goal. We will clarify these objectives in the revised manuscript to underline that we cannot (and do not)*

145 *claim that the current version of the method can be considered fully validated.*

L. 145–149: At the first reading, this part was hard to understand, it only became clear after reading the appendix. It is not fully correct either, the "distributions" are nowhere shown, only the median and the percentiles are shown in the appendix. Please revise for readability, and include all the necessary information, the median/percentile of what exactly? "2nd and 3rd order polynomial" could be stated explicitly instead of vaguely referring to "a polynomial function". Does the "integrated

150 energy flux" take into account the different electron masses between SSJ and eVlasiator? See also general comment #1. And what is finally done with the adjusted ratio, are the eVlasiator fluxes simply multiplied/divided by that factor? Please be more specific about how the final values are obtained.

*Thank you for pointing out that this part is difficult to understand without reading the appendix. We will revise the text to better guide the reader.*

155 *We will also add the requested elements to ease the understanding and the improve the accuracy of the description of the scaling process.*

L. 167: What altitude is "above its altitude range"? How can WACCM account for any effect above the altitude range? Although NOEM is derived for up to 200 km, the main NO distribution is centred around 105–110 km. How is this combined with the auroral forcing, both from the "normal" 2 keV "Kp-model" and the eVlasiator input to avoid double counting? Please

160 be more specific about how the individual sources are combined.

NOEM is used to set the upper boundary condition of NO concentration in WACCM simulations. This accounts for the production of NO at altitudes not covered by the WACCM domain, and it's exclusion would lead to a lack of NO at high altitudes. NOEM is a standard component of WACCM simulations, and we are not aware of any discrepancies arising from its use as the NO upper boundary condition. In our revised analysis we have also limited the energy range of the eVlasiator-derived auroral electron fluxes to energies greater than 100 eV. This prevents overlap with NOEM, as electrons at energies below 100 eV would precipitate at altitudes above 140 km.

The use of NOEM in WACCM makes it necessary to set the $Kp$ value for the VLAS simulation, as explained in Section 2.4.2 of the manuscript. We will clarify this further in the revised manuscript.

L. 175: See general comment #1, does this electron energy take into account the changed electron mass? How is it converted to the real electron mass used and energy used as input for WACCM?

The electron energies are calculated using the eVlasiator electron mass. Given the reduced mass ratio constraint of eVlasiator, we obtain the electron differential number flux in terms of energy, for the eVlasiator output. We assume the energisation of these high-mass electrons in eVlasiator is still representative. We take this differential number flux $\mathcal{F}_e(E, \mathbf{r})$ of eVlasiator electrons and apply it as the input of WACCM as-is, assuming the diff. number flux of real-mass electrons has the same distribution in energy as the eVlasiator output. We now state this explicitly in the manuscript.

L. 176: The parametrization in [5] was derived for an electron energy range from 100 eV to 1 MeV, How did the authors extend that range to lower energies of 50 eV? Is the energy grid in log-space, i.e. in log(energy)?

Thank you for pointing this out. Indeed, spectral energy range of 50 eV to 50 keV is used in the ionisation rate calculation. The energy grid has 32 log-spaced points most of which can be seen in Fig. 2. Three of these energy grid points are below 100 eV. We have now removed those three points and calculated the atmospheric ionisation rates again. At the model altitude range considered here, i.e. below 140 km, the removal of <100 eV electrons from the spectrum does not change ionisation. As pointed out by another Reviewer, electrons with such low energy cannot penetrate below the model's upper altitude limit. We have also re-examined the ionisation rates at the higher end of the auroral electron energy spectrum. Since the medium-energy electron (MEE) forcing recommended by CMIP6, which was used in our WACCM simulations, already accounts for electron forcing at energies greater than 30 keV, we have also removed the eVlasiator derived auroral electron fluxes at energies 30–50 keV. This avoids the possibility that the eVlasiator auroral forcing and the MEE forcing might overlap. The eVlasiator derived electron fluxes at these energies were so small, that this removal also had no considerable impact on the results of our study.

The final energy range of the eVlasiator-derived auroral electron input used in WACCM is 100 eV – 30 keV. As explained above, the adjustment of the auroral electron precipitation spectral energy range has negligible impact on the calculated ionisation rates and corresponding atmospheric response. We will add a brief discussion about this issue to Section 2.4.1 of the manuscript.

L. 177: Why was NRLMSIS-00 used for the ionization rates, and not the WACCM atmosphere? NRLMSIS-00 provides a good climatology, but here it might be a good idea to use WACCM itself, especially in terms of consistency. This choice needs to be motivated better, especially since CMIP6 is not part of the presented paper, and the method described in [6] and [7] is used for medium-energy electrons.

The ionisation rate calculation requires an atmosphere which for the calculation was taken from the NRLMSISE-00 model. However, according to the CMIP6 procedure (Matthes et al. 2017), the ionization rates are then divided by the MSIS mass density which effectively removes the atmospheric "signature". When the rates are used in WACCM, they are multiplied by the WACCM mass density profiles making the forcing consistent with the WACCM atmosphere.

We will note this in Section 2.4.1 of the manuscript.

L. 186: "WACCM-D runs from ... 2005 to ... 2006" is confusing and needs more explanation. The study compares the impact of events in 2011 and 2015, yet the WACCM runs are for an entirely different time period. What is the purpose then, and how do the authors reconcile consistency?

205   The primary goal of our study is to present a new methodology for the evaluation of the atmospheric impacts of precipitating auroral electrons. We are aware that the method currently suffers from many limitations, including the use of only a single set of auroral electron fluxes derived from eVlasiator. As such, we are not interested in the evaluation of any specific event, rather the type of forcing. Indeed this is the reason we use observations from two distinct periods to scale the eVlasiator electron fluxes, to mitigate the strong variability exhibited by real-life particle precipitation.

210   We therefore argue that the choice of the simulation period for the WACCM runs is of lesser importance. The fact that for each WACCM simulation the auroral electron forcing is kept constant already presents an unrealistic premise, but is in line with our goal of evaluating the type of forcing of the atmosphere. Since all the WACCM simulations include the same components, with the exception of the auroral electron forcing, they are comparable to each other, and differences between the runs arise from differences in the precipitating auroral electron data.

215   We will add a brief discussion on the choice of the WACCM simulation period to the manuscript in the revision.

L. 190: "turns off Kp ...", what about the NOEM input which is also aurora related? Is it switched off as well for the "no-aurora" reference run?

NOEM is still included in the VLAS and REF simulations. Since the ionization rates derived from eVlasiator can only affect the NOx production within the altitude range of WACCM, NOEM is necessary to set the upper boundary NO of the WACCM

220   simulations. This ensures comparability with the $Kp$-driven auroral forcing, while excluding NOEM results in a lack of NOx at high altitudes in WACCM. We will make this clearer in the revised manuscript.

L. 197: "... no Kp-driven ... aurora" depends on if NOEM has been switched off or not, otherwise there will be remnants of Kp-driven aurora via NOEM.

As NOEM is necessary in WACCM simulations to set the upper boundary of NO concentration, it is true that some auroral

225   electron forcing effect is present in the REF simulation. We will clarify in the revised manuscript that NOEM is still included in the REF simulation, as this ensures comparability of the WACCM simulations.

L. 203: For completely "no aurora", NOEM should be switched off too. It is not clear from the text.

NOEM is included in all our WACCM simulations. We will clarify this explicitly in the revised manuscript.

L. 206: Why is Kp 2 needed for eVlasiator? Shouldn't it drive all the aurora itself? What Kp-level would correspond to the

230   solar wind conditions used in the Vlasiator simulations? Table 1: The difference between REF and Kp0 could be made a little clearer.

It is necessary to set the $Kp$ index value for the VLAS run, since it is used by NOEM to set the NO upper boundary condition. The auroral electron forcing derived from eVlasiator therefore only accounts for the ionization occurring withing WACCM's altitude range. Processes above/at the top of WACCM's altitude range are accounted for through the NO concentration set by

235   NOEM at WACCM's upper boundary.
We have chosen the $Kp$ index for the VLAS simulation based on the $Kp$ index during the DMSP satellite observations used to scale the eVlasiator electron fluxes.
The difference between the REF and KP0 simulations is the inclusion of the $Kp$-driven parameterization of the auroral electron forcing. The REF run uses the same modification as the VLAS run to disable the Kp parameterization, while the KP0

240 includes the parameterization. As can be seen in Fig. 4b, while the parameterized ionization rates may be low, setting the $Kp$ index to 0 does still produce some auroral electron forcing.

We will clarify these aspects of the WACCM simulations in the revised manuscript.

L. 214: Does the "precipitating electron integrated energy flux" account for the heavier electron mass?

As stated in earlier replies, the precipitating electron integrated energy flux calculation uses the real electron mass.

245 L. 222: Please state the altitude range of the vertical integration.

The ionization rates in Fig. 4 are integrated from the surface to the top of the model (140 km). Since the auroral electron forcing only occurs at the highest altitudes, much of this altitude range does not contribute to the total integral. We will state this in the revised manuscript.

L. 223: The described features are very difficult to see with the chosen colourmap. See also below my comment and
250 suggestion for this figure.

We will be updating Fig. 4, including the colourmap, during the revision.

L. 227: The limited magnetospheric domain only explains the hard cutoff at the lower latitudes. The VLAS ionization rates also do not reach as high latitudes as the KP2 run, between $10^{-4}$ and $10^{-5}$ hPa. What would be the reason for this?

The poleward edge of the precipitation corresponds to the open-closed field line boundary, which is captured by (e)Vlasiator
255 at a specific point in time, and this is realistic (see, e.g. Newell et al., 1996, doi:10.1029/95JA03516). In contrast, statistical models contain contributions from varying conditions, blurring the otherwise-sharp poleward edge.

L. 233: I would assume that the "lower boundary" is because of the fixed 2 keV energy for the Kp-WACCM, and probably the limited altitude range used to derive it.

Yes, this is likely to be the case.

260 L. 237: Has the "polar cap average" been calculated with area weighting (cos (lat) weighting)? Please state it if so, otherwise not weighting correctly will over-emphasise the low values at high latitudes.

The polar averages have been calculated using cos-weighting to account for the differences in area represented by grid points at different latitudes. We now state this in the revised manuscript.

L. 240: See general comment #3, please add the absolute changes for reference.

265 We will add absolute values of the changes to the revised manuscript.

L. 249: See comment above, why is Kp needed for the Vlasiator run?

As stated in previous responses, it is necessary to set the $Kp$ index value for the VLAS run, since it is used by NOEM. NOEM sets upper boundary condition of NO concentration in WACCM.

L. 254: As the authors point out, there was a NH SSW happening in 2006, which makes it even more confusing why the authors have chosen 2005 and 2006 for the WACCM simulations, and not 2011 or 2015 as for the comparison with SSJ.

As stated previously, our approach was specifically not to evaluate any specific events, and the use of constant auroral forcing throughout the WACCM simulation period would have made such assessments unrealistic. Since the polar vortex is generally known to be more unstable in the NH than the SH, this likely would have created differences between the hemispheric winters no matter the selected simulation period.

L. 258: The authors could show the polar-cap average time-altitude distribution in both hemispheres showing the downward transport for comparison with earlier studies.

Thank you for the suggestion. We will consider adding a figure depicting the NOx descent during the revision of the manuscript.

L. 261: I could not find any profiles presented as line plots. From the maps it is difficult to figure out what the authors are discussing here. Please add at least one figure actually showing profiles for a visual comparison.

Thank you for pointing out that the text and Fig. 7 do not adequately correspond to each other. We will revise the text and consider adding a figure specifically depicting NOx profiles to the revised manuscript.

L. 268: How would "solar irradiation" work in polar winter?

Thank you for pointing out this oversight. NOx production during the polar winter is driven by geomagnetic activity, including energetic particle precipitation. We will correct this in the revised manuscript.

L. 271: "... adds to the effect." What effect? It is unclear what the authors are trying to point out here.

This refers to the off-pole nature of the distribution of NOx in the NH due to the weaker polar vortex. Since SSW leads to a less stable polar vortex, the SSW also leads to the less symmetrical distribution of NOx in the NH (Fig. 7d–f) compared to the SH (Fig. 7a–c). We will clarify this in the revision.

L. 277–278: "higher REF levels of NOx", higher than what? Please clarify.

Thank you for pointing out this ambiguous language. This is in reference to the difference in the lower thermospheric NOx concentrations in the SH (Fig. 7a) and NH (Fig. 7d) in the REF simulation. The difference in REF NOx concentrations between the hemispheres helps explain the differences in the VLAS impacts in Fig 7g,j. We will clarify this in the revised manuscript.

L. 280: "difference in the REF background levels", difference between what? Please clarify.

As above, this is in reference to the differences in the REF NOx concentrations between the two hemispheres, the SH (Fig. 7b) and NH (Fig. 7e). We will clarify this in the revised manuscript.

L. 293–298: I suggest to move these sentences to the conclusions since they provide more of a summary than a discussion.

We will make the suggested changes in the revision.

300    L. 303: Add "in WACCM": "current parametrization of aurora in WACCM ...". How do the authors arrive at the conclusion that the "standard" auroral forcing overestimates it? Overestimation compared to what? So far, no comparison to NOx measurements is presented.

We will add "in WACCM" in the revision. The assessment of the overestimation was made based on the parametrised ionisation rates on average being higher than the ionisation rates derived from the DMSP-scaled eVlasiator electron fluxes. It
305    is true that further validation is needed for this statement, which is why the sentence is phrased as a hypothetical in the manuscript. We will reconsider this wording and its justification in the revision.

L. 310: What makes the authors so sure that they are "underestimated"?

As eVlasiator does not include all sources that are known to produce precipitating auroral electrons, the electron fluxes are expected to be lower than the total, but it is true that this might not be the case. As with line 129, we suggest to rephrase this
310    and the following sentence as follows: "As pointed out in Sect. 2.3.1, eVlasiator does not model all sources of precipitating auroral electrons, and therefore the obtained electron fluxes might differ from reality. We have mitigated the possible discrepancy in this study by using the DMSP observations to scale the electron fluxes."

L. 323–325: See general comment #3, please provide absolute changes as well.

We will add the absolute changes to the text.

315    L. 333: I suggest to replace "made" by "carried out".

We will make the suggested changes in the revision.

L. 336: I suggest to replace "makes way" by "paves the way".

We will make the suggested changes in the revision.

L. 350: I might be nitpicking here, but since a field line is an integral curve of the vector field (axial vector field in the case of
320    **B**), it is already "traced". Thus "tracing a field line" seems tautological, and "tracing the field" or "field lines are calculated" would already be unambiguous enough.

We will replace "field lines are traced" by "field lines are followed" in the revised version of the text.

L. 351: I suggest to use the term "initial point" instead of "seed point", as I assume that is what the authors are describing.

We will replace "seed point" with "start point" in the revised manuscript, following a suggestion from another Reviewer.

325    L. 359: "follow the geomagnetic field" is unambiguous enough. But again, what magnetic field model was employed here?

We will modify the phrasing as suggested in the revision. As stated in the sentence, the magnetic field model employed here is the Tsyganenko 2001 model to account for the external magnetic field contributions, combined with the untilted dipole accounting for the internal contributions. More details can be found in the referenced paper (Grandin et al., 2023).

L. 365: "Better self-consistency" feels a bit strange, better in what sense? In my opinion, things either are consistent or they are not.

Thank you for pointing this out. In the revision, we will replace this phrase with "to be consistent".

L. 365: I suggest to remove "lines" and "file" here, it is not clear what this "file" is and how it can be obtained.

We will make the suggested changes in the revision.

L. 368: It might just be a poor choice of words, but to my understanding a field line can only end on a surface, not in free space except when the (magnetic) field becomes exactly zero. Please revise for clarity.

We will modify the phrasing as follows in the revision: "and either the point where the magnetic field tracing was stopped (point $C_2$)".

L. 369: I suggest to remove "therefore".

We will remove "therefore" in the revision.

L. 385: The distinction could also refer to NH and SH instead of dayside and nightside, since the dayside passes are in the NH, and the nightside passes are in the SH. Only an additional comparison during NH nighttime and SH daytime would allow to use one or the other.

In this section, we are interested in scaling the precipitating fluxes on the dayside and on the nightside separately, as the dependence of auroral electron fluxes is predominantly related to magnetic local time. While interhemispheric asymmetries could be studied if suitable satellite overpasses were available to measure the particle fluxes in both hemispheres at the same MLT, they can be expected to be marginal in comparison to the dayside-vs-nightside differences. Therefore, we believe that referring to NH and SH in this paragraph would misportray the approach and further confuse the reader.

L. 388–391: A figure marking the selections for Oct 2015 is missing, in addition to Figs. B1–B3.

Since the purpose of Fig. B3 is solely to illustrate the MLAT–MLT domains retained as "regions of interest" for both events, we believe that adding the version with the data of the 10 October 2015 event in background would not bring more information and would only contribute to making the appendix harder to digest for the reader. We will however add a sentence clarifying that the regions of interest apply to both events in the revised text.

L. 392–393: This sentence essentially repeats the statement above it and can be removed.

Thank you for pointing this out; we will remove this sentence in the revision.

L. 395–408 (Appendix B3): This appendix describes a method that is not used in the end and should be removed to reduce confusion, including Fig B4. A (more extensive) comment about the issue with mean/median can be included in appendix B4. However, a better description is needed about over which dimension (of Fig. B3e,f) the quantiles are calculated. I can only assume that the authors mean "along the DMSP orbit", or equally "time". If that is correct, please state it in the text to avoid too much guessing there. Since the percentile is presented on a log-log plot, how do the mean and median of the log-log

behave? Are the distributions still skewed as in the linear case? Are the correction factors applied to the eVlasiator fluxes or to their logarithm?

Thank you for the suggestion to remove Appendix B3. However, we strongly believe that the scaling method, already difficult enough to follow for the reader, would become even more confusing if we were not to first introduce the reasoning by taking the example of using the median values. If we were to state from the beginning that we use the 61st and 67th percentiles to apply the scaling, those numbers would seem completely arbitrary. This is why we first explain how the energy-dependent correction is designed on differential number fluxes, to only later evaluate whether the correction was sufficient in the sense of integrated energy fluxes. We will clarify in the revision that the percentile values are indeed calculated along the DMSP orbits. The correction factors in their polynomial form are converted back to the linear domain before being applied to the eVlasiator fluxes; this will be explicitly stated in the revision.

We are unsure as to what the Reviewer means by "Since the percentile is presented on a log-log plot, how do the mean and median of the log-log behave? Are the distributions still skewed as in the linear case?"

L. 410–414: This description is very hard to understand, please revise. The 0.9 quantile of what exactly, and why should its ratio be 1?

Thank you for pointing out that this paragraph is not clear; we will rephrase and expand it in the revision. In particular, we will be more precise in explaining which parameter we consider the 90th percentile of and the reasoning behind the choices.

L. 411: "Insufficient" for what?

We will add the following precision in the revised text: "However, this increase is still insufficient to be representative of the energy input into the upper atmosphere associated with auroral electron precipitation as obtained in the DMSP observations".

L. 412: Referring to my earlier remarks, does this comparison of (integrated) energy fluxes account for the different electron masses used in eVlasiator and DMSP/SSJ?

Please see the earlier replies on this subject.

L. 410–419 and main text, either replace "quantile" with "percentile" or adapt the numbers to real quantiles between 0 and 1 (divide by 100).

Thank you for pointing out this language issue; we will replace "quantile" with "percentile" throughout the revised manuscript.

L. 418: What is the final ratio of the integrated energy fluxes?

The obtained ratios of the integrated energy fluxes are 0.99 and 1.03 for the cusp and nightside regions, respectively. We will add this information in the revision.

L. 422, Figs. B4, B5: The authors fit a third-order polynomial in log(energy), based on an energy range from 50 eV to about 5 keV (hard to identify in the figures). However, the "final" energy range is from 50 eV to 50 keV, and the 3rd order polynomial reaches a correction factor of about $10^5$ already at 10 keV. Is there any "safety" mechanism that keeps the correction factor in a sensible range? Using an empirical model (here the polynomial) outside the parameter range used for fitting is dangerous, unless the authors can ensure that their model is valid outside that range.

There is no such risk, given that the high energies for which the ratio (blue line) is not defined are those for which the eVlasiator fluxes are zero. Therefore, whichever the value of the polynomial at those energies, the corrected fluxes will still be zero. Thank you for pointing this out, though, as it is something that we will clarify in the revision.

Figures
Fig. 4: I suggest to find a better colourmap, the current one makes it very difficult to distinguish VLAS from KP0–KP2. Maybe overlay contour lines from VLAS over all panels for comparison.

We will be updating Fig. 4 for the revised manuscript, including the colourmap and scale. To further highlight the differences between the eVlasiator-derived and the $Kp$-driven auroral electron forcing, we will also include momentary snapshots of the auroral ionisation rates, in addition to the currently shown daily averages.

Fig. 5: A suggestion would be to show a geomagnetically averaged zonal mean instead of a fixed longitude. For direct comparison, one could add the contours from one panel to the other.

Thank you for the suggestions. We will consider these during the revision of the manuscript.

Fig. 6: Please replace the bright yellow line by a different colour, it is very difficult to see on white paper. Please add the altitude ranges in to the panels or their titles so that they are easier to identify. The authors might consider removing the KP5 line so that the y-axis scale can be reduced, making the other lines easier to distinguish. Again, showing only relative differences is only part of the story, an additional comparison of absolute values would be helpful. And if the authors can find, real NO data would be very interesting to compare to as well.

We have updated the colours used in Figs. 6 and 8, and added the approximate altitude ranges to panel titles. The KP5 simulation has been replaced with KP3 in Figs. 6 and 8, and the scales have been adjusted accordingly. We will add the absolute NOx changes to the text in the revised manuscript.

Fig. 7: Please revise the colourmap of the panels (a)-(f), the features described in the text are difficult to see in the figure. In particular the black contour lines and labels are unreadable.

We will adjust the colourmap used in Fig. 7a–f to make it clearer in the revised manuscript.

Fig. 8: See comment about Fig. 6 and replace the bright yellow line by a different colour. Similar, reducing the Kp range to 0–2, the y-axis scale can be adapted to better distinguish the other lines. Also, please add the absolute changes for reference. Please label the panels clearly with "SH" for (a) and "NH" for (b) to make that easier to find.

We have updated the colours used in Figs. 6 and 8 and added the approximate altitude ranges to panel titles. The KP5 simulation has been replaced with KP3 in Figs. 6 and 8, and the scales have been adjusted accordingly. The hemispheres have been added to Fig. 8. We will add the absolute O3 changes to the text in the revised manuscript.

Fig. B5: Instead of only showing a line for the selected percentile, the authors could add contours for the underlying distributions, including additional lines for the mean and the median, with an emphasis on the percentile line that was used in the end. This would make it clear how skewed these distributions are, since the authors claim that using the mean or median are not good choices.

Thank you for this suggestion; we will consider it when preparing the revision of the manuscript.

References

[1] Alho, M. et al. (2022), Electron Signatures of Reconnection in a Global eVlasiator Simulation, Geophys. Res. Lett., 49(14). doi:10.1029/2022gl098329.

[2] Wissing, J.M. and Kallenrode, M.-B. (2009), Atmospheric Ionization Module Osnabrück (AIMOS): A 3-D model to determine atmospheric ionization by energetic charged particles from different populations, J. Geophys. Res. Space Phys., 114(A6), A06104. doi:10.1029/2008ja013884.

[3] Tyssøy, H.N. et al. (2021), HEPPA III Intercomparison Experiment on Electron Precipitation Impacts: 1. Estimated Ionization Rates During a Geomagnetic Active Period in April 2010, J. Geophys. Res. Space Phys., 126, e2021JA029128. doi:10.1029/2021ja029128.

[4] Sinnhuber, M. et al. (2021), Heppa III Intercomparison Experiment on Electron Precipitation Impacts: 2. Model-Measurement Intercomparison of Nitric Oxide (NO) During a Geomagnetic Storm in April 2010, J. Geophys. Res. Space Phys., 126, e2021JA029466. doi:10.1029/2021ja029466.

[5] Fang, X. et al. (2010), Parameterization of monoenergetic electron impact ionization, Geophys. Res. Lett., 37(22), L22106. doi:10.1029/2010gl045406.

[6] Kamp, M. van de et al. (2016), A model providing long-term data sets of energetic electron precipitation during geomagnetic storms, J. Geophys. Res. Atmos., 121(20), 12, 520–12, 540. doi:10.1002/2015jd024212.

[7] Matthes, K. et al. (2017), Solar forcing for CMIP6 (v3.2), Geosci. Model Dev., 10, 2247–2302. doi:10.5194/gmd-10-2247-2017.

---

## Author Response (AR2)

**Authors' response to referee comments on "Atmospheric odd nitrogen response to electron forcing from a 6D magnetospheric hybrid-kinetic simulation" by Häkkilä et al.**

The authors would like to thank all the Referees for the review process. We are grateful for the Referees' insightful and constructive comments and we appreciate the time taken to review the revised manuscript. We also appreciate the Referees' acknowledgements of the improvements made to the manuscript in the revision.

Please find below our answers (in blue) to the comments (in black).

**Response to the comments of Referee #1**

I'd like to thank the authors for responding constructively to the comments from the other reviewers and myself. I think the manuscript has improved considerably through this and I don't have major concerns regarding the publication anymore.

10

5

I have to admit, that I'm only halfway satisfied with the replies to my original first major concern, the vagueness of the motivation. I understand that the manuscript presents a novel approach. But when the authors write about "considerable uncertainties in our knowledge of the polar MLTI". I would like to see these uncertainties to be spelled out and argued why this novel approach has the potential to reduce these uncertainties. There is the helpful reference to Sarris et al. (2023) on

- 15 questions concerning plasma-neutral interactions, and in the paragraphs following this statement it is becoming clear that this study has a focus on NOx production from auroral electrons which are unsatisfactorily represented in existing approaches. I also appreciate the argument that "solar wind parameters can be observed earlier than e.g. the geomagnetic activity" provided newly in the Discussion section. However, I still think the authors miss the chance of providing a clear statement on the potential of their new approach to answer existing questions. Which questions would that be?
- 20 Thank you for pointing out that the motivation of the study still remained unclear. As the Reviewer states, we are focused on the production of NOx from auroral electron precipitation, and how this precipitation is presented in atmospheric modelling. In addition to presenting a new method of deriving the auroral electron precipitation, we focus on whether this more accurate characterisation of auroral electron forcing could solve the inadequate production of NOx in current atmospheric models. We now state this motivation more explicitly in the Introduction of the manuscript, and have modified the Discussion section
- 25 accordingly.

**Response to the comments of Referee #3**

The manuscript presents a novel approach to combine magnetospheric modelling with whole-atmosphere modelling to investigate the auroral production of odd nitrogen. The magnetospheric model drives the precipitating electrons, and the whole-atmosphere model WACCM is used for the production of the odd nitrogen species, NO in this case.

30

The authors have addressed all my concerns and clarified the respective points which has improved the manuscript considerably. Thus it is a worthy contribution to Ann. Geophys., I have only a few minor points left that should be addressed before publication.

35 General comments

1. I would still like a short note about the single WACCM model run used in the analysis. Even though the authors refer to the "SD" part, the interesting part is where the model is free-running. It is not clear or even obvious that multiple runs will produce identical results as the authors suggest.

40 Thank you for the comment. We have now included a brief discussion on the use of single WACCM runs in this study in the Discussion section of the manuscript.

2. Appendix B4: To my understanding, the equations given by the authors describe the integrated particle flux [cm-2 s-1 sr-1], not the integrated energy flux [eV cm-2 s-1 sr-1]. I assume both can be used for the normalization here, but I suggest to correct the terminology.

45 Thank you for pointing this out. We did use the integrated energy fluxes for the described purpose, and indeed forgot the energy term in the integral expressions. We have corrected the equations accordingly.

Specific comments

- L. 25: Maybe "complex interactions" instead of "dynamics"?
- 50 We have made this change in the revised manuscript.

L. 114–115: A short note about how typical these solar wind conditions are would be helpful.

We have added a brief mention to such conditions being found during the fastest of the solar wind high-speed streams and how frequently such events typically occur.

L. 124–127: This description is a bit long and convoluted with repeated expressions. I suggest to rewrite to be a bit more concise.

We have rephrased these sentences in a more condensed and hopefully clearer way.

L. 193: How many model levels are used for NOEM input?

NOEM is used to set the NO concentration at the top-most model level of WACCM, i.e. at one level.

L. 302–310: The authors list only the numbers of the SH, please add the NH relative and absolute changes too. That would
strengthen the authors' argument later in the manuscript, that in the NH the background levels are higher, so that the relative changes are smaller.

We have now included more NH  $NO_x$  impact numbers, both relative and absolute changes, in Section 3.2 when discussing the magnitude of the impact. For some specific examples of peak impacts we still only include the SH values.

L. 364–370: The authors list only the numbers of the SH, please add the NH relative and absolute changes too.

65 We have now included more NH ozone impact numbers, both relative and absolute changes, in Section 3.2 when discussing the magnitude of the impact.